# Towards Persistent Noise-Tolerant Active Learning of Regular Languages with Class Query

**Lekai Chen, Ashutosh Trivedi, and Alvaro Velasquez**
Department of Computer Science
University of Colorado, Boulder
Boulder, CO 80309, USA
`{lekai.chen,ashutosh.trivedi,alvaro.velasquez}@colorado.edu`

## Abstract

Large Language Models (LLMs) are increasingly deployed in human-AI collaborative decision-making systems, where they are expected to align precise formal representations (e.g. temporal-logic monitors or reward machines for reinforcement learning) with ambiguous natural language. However, their ad hoc strategies for resolving ambiguity often lead to hallucinations and inconsistencies. We formalize this setting via probabilistic Minimally Adequate Teachers (pMATs) that (i) answer membership queries with fixed but possibly flipped labels, and (ii) return valid counterexamples to hypothesis equivalence. We present CAPAL (**C**lass-query **A**ctive, **P**ersistent-noise-**A**ware **L**earning), an active learning algorithm for learning deterministic finite automata (DFAs) without demonstrations and that remains correct under persistent noise in membership queries. CAPAL augments the classic $L^\star$ loop with two components grounded in our implementation: (1) a *class query* realized as a statistical same-state test that compares disagreements between two prefixes against a noise-floor estimate $\hat{\eta}$ with Hoeffding tolerances; (2) a *discrimination tree (DT)* that selects a near-minimal discriminator, keeping the core suffix set small. An efficient micro-bootstrap and cache-reuse scheme estimates $\hat{\eta}$ with few new queries. We prove convergence given a perfect language-equivalence oracle and show substantial membership-query savings in practice. Our evaluation across multiple benchmarks, including RegexLib and KB13, demonstrates that this approach enhances both the efficiency and robustness of DFA learning under noisy oracles, supporting the view of LLMs as fallible yet useful collaborators for synthesizing verifiable formal artifacts.

## 1 Introduction

Formal languages offer a principled framework for modeling human preferences and values in a precise, verifiable, and interpretable manner, making them ideal for aligning the behavior of neural networks with high-level human intent. In reinforcement learning, temporal logics such as LTL have been used to specify complex tasks (Camacho et al., 2019; Voloshin et al., 2022; Bagatella et al., 2025), while automata-based representations like reward machines enable modular, structured reward functions (Toro Icarte et al., 2019; 2022; Camacho et al., 2019). Safety-critical systems benefit from shielding techniques, which enforce logical constraints at runtime to prevent unsafe actions (Alshiekh et al., 2018). In neuro-symbolic systems, logical rules have been incorporated into the training process through logic regularization or differentiable programming, allowing human-aligned constraints such as fairness to shape learned model behavior (Innes & Ramamoorthy, 2020). These developments demonstrate that formal artifacts are not only expressive enough to capture the richness of human values, but are also actionable and integrable within neural architectures for AI alignment. In this paper, we study how such formal representations—regular languages and their automaton realizations—can be learned via interaction with a (noisy) large language model (LLM) under a probabilistic extension of Angluin's Minimally Adequate Teacher (MAT) framework (Angluin, 1987).

Constructing such formal artifacts by hand is cumbersome and error-prone, especially when only informal or scattered specifications exist. Recent work uses LLMs to close this gap by extracting automaton-based task knowledge from natural language and integrating the resulting automata into planning and verification pipelines (Yang et al., 2025c;b;a). As a concrete example, consider an autonomous driving agent deployed in a new city whose driving conventions differ from those seen during training: a designer specifies desired behavior in natural language, an LLM judges whether example trajectories satisfy these guidelines, and a driving simulator tests candidate automata and supplies counterexample trajectories. The learned automaton then serves as a reward machine or safety shield for reinforcement learning, with the LLM acting as a noisy membership oracle and the simulator providing equivalence queries by searching for disagreeing trajectories—precisely the probabilistic MAT (pMAT) scenario we formalize below.

Regular languages occupy a well-behaved corner of the Chomsky hierarchy, with rich closure properties, canonical representations, and efficient algorithms for manipulation and inference. Among their many descriptions, DFAs stand out for interpretability, expressiveness, and learnability; they have consequently become central to formal alignment (Toro Icarte et al., 2019; 2022; Camacho et al., 2019; Bejerano, 2003; Liu et al., 2023; Merrill, 2019).

Active learning of DFAs, epitomized by $L^*$ and its descendants (e.g., TTT) (Angluin, 1987; Kearns & Vazirani, 1994; Rivest & Schapire, 1989; Denis et al., 2004; Bongard et al., 2005; Isberner et al., 2014; Volpato & Tretmans, 2015), proceeds by querying an oracle with *membership queries* (MQs) and *equivalence queries* (EQs) to iteratively refine a hypothesis until it matches the target language. However, deploying LLMs as oracles raises robustness concerns. Prior work has explored two main uses of LLMs in automata inference: (i) employing them as noisy MQ oracles (Vazquez-Chanlatte et al., 2024), often requiring human demonstrations to bootstrap the learner; and (ii) prompting them to synthesize DFA transitions directly from natural language (Alsadat et al., 2024), implicitly assuming reliable translation from informal specifications to precise transition structures. Neither route provides correctness guarantees because hallucinated outputs are difficult to detect and correct systematically.

Passive DFA learning methods, such as RPNI and its variants (e.g., RPNI-EDSM Lang et al. (1998)), infer automata from fixed sets of labeled examples using state-merging techniques Oncina & García (1993); Lang et al. (1998). While these algorithms are PAC-learnable under certain conditions Board & Pitt (1992), their performance is often constrained by limited data and overfitting risks Bugalho & Oliveira (2005); Board & Pitt (1992). Enhancements like combining passive learning with active strategies Yang et al. (2019) offer improvements but do not address the persistent errors in membership queries that challenge real-world applicability.

The broader challenge of learning with oracle errors is well known. Angluin et al. (1997) showed that DFAs remain learnable under finitely many MQ errors when the EQ oracle is perfect, via the LEARNANYWAY algorithm, which replaces suspected mistakes with counterexamples. In practice, however, short counterexamples are rarely available (Isberner et al., 2014). Active learning algorithms decide what MQs to ask vy have to analyzing the CE. As CEs grow in length, the number of LLM-induced mislabels encountered during refinement can scale superlinearly, rendering naïve applications of LEARNANYWAY impractical.

To address these limitations, we introduce the *pMAT* formalism, in which the oracle may persistently err on MQs but always returns valid counterexamples of arbitrary length when EQs fail. Building on this model, we propose CAPAL (**C**lass-query **A**ctive, **P**ersistent-noise-**A**ware **L**earning), an active DFA learning algorithm that remains correct under persistent membership noise and requires no demonstrations. CAPAL leverages two key ideas: (i) even a noisy membership oracle can be exploited statistically to decide whether two prefixes belong to the same Myhill–Nerode class (a *class query*); and (ii) discrimination trees enable efficient hypothesis refinement with near-minimal distinguishing suffixes (Isberner et al., 2014). To further strengthen robustness, we augment the framework with an *adaptive reasoning pipeline* that allows LLMs to invoke external symbolic solvers or simulators when processing MQs, improving both accuracy and practical applicability.

Our main contributions are:

1. We introduce **pMAT**, an extension of Angluin's MAT that models noisy LLM membership responses while preserving exact counterexamples from the equivalence oracle.

Figure 1: LLMs as Probabilistic MATs.

2. We formalize the notion of *persistent errors*, showing why mislabels produced by LLMs as pMATs cannot be corrected by re-sampling and must be resolved through counterexamples.

3. We propose CAPAL, the first active DFA learning algorithm that provides *theoretical correctness guarantees under persistent MQ noise*, requiring neither demonstrations nor auxiliary supervision.

4. We design a *code-based oracle* that leverages LLMs to synthesize deterministic checkers, converting natural-language specifications into executable predicates that answer MQs consistently and reliably.

## 2 PRELIMINARIES

**Deterministic Finite Automata.** Let $\Sigma$ be a finite set of symbols called the alphabet. A language over $\Sigma$ is a set of words (i.e., finite sequences of symbols from $\Sigma$). A regular language is one that is representable by a finite automaton.

**Definition 2.1** (DFA). *A deterministic finite automaton (DFA) over an alphabet $\Sigma$ is a tuple $\mathcal{A} = \langle Q, \Sigma, q_0, \delta, F \rangle$ where $Q$ is a finite set of states, $q_0 \in Q$ the initial state, $\delta : Q \times \Sigma \to Q$ the transition function, and $F \subseteq Q$ the set of accepting states.*

The transition function extends to words in the standard way: $(\delta^{\mathcal{A}})^* : Q^{\mathcal{A}} \times \Sigma^* \to Q^{\mathcal{A}}$. The language of $\mathcal{A}$ is defined as $\mathcal{L}(\mathcal{A}) = \{ w \in \Sigma^* : (\delta^{\mathcal{A}})^*(q_0^{\mathcal{A}}, w) \in F^{\mathcal{A}} \}$. For a word $w \in \Sigma^*$ we use the shorthand

$$\mathcal{A}[w] := (\delta^{\mathcal{A}})^*(q_0^{\mathcal{A}}, w) \in Q^{\mathcal{A}}, \qquad \mathcal{A}(w) := \begin{cases} 1 & \text{if } \mathcal{A}[w] \in F^{\mathcal{A}}, \\ 0 & \text{otherwise.} \end{cases}$$

Thus $\mathcal{A}[w]$ denotes the reached state, whereas $\mathcal{A}(w)$ denotes the acceptance label.

**Learning DFA from MATs** In Angluin's minimally adequate teacher (MAT) framework (Angluin et al., 1997), a learner interacts with an unknown target DFA $\mathcal{M}$ only through two types of queries.

*MQs* ask, for a word $w \in \Sigma^*$, whether $w$ belongs to the target language: the teacher returns $\mathcal{M}(w) \in \{0, 1\}$.[1] *EQs* present a hypothesis DFA $\widehat{\mathcal{M}}$; the teacher either answers that $\widehat{\mathcal{M}}$ is correct or provides a counterexample word $w$ with $\widehat{\mathcal{M}}(w) \neq \mathcal{M}(w)$. We assume the learner stores all MQ and EQ interactions in caches $C_{MQ}$ and $C_{EQ}$, and write $C_{MQ}(w)$ for the (possibly noisy) label stored for $w$ when defined.

The $L^*$ algorithm is a canonical polynomial-time learner in this setting. It maintains an *observation table* with a finite set $S \subseteq \Sigma^*$ of prefixes (including $\varepsilon$) and a finite set $E \subseteq \Sigma^*$ of suffixes. For each $s \in S \cup S\Sigma$ and $e \in E$, the learner issues an MQ on the concatenation $se$ and records the answer

---

[1] Here $\mathcal{M}(w) = 1$ iff $w \in \mathcal{L}(\mathcal{M})$.

$\mathcal{M}(se)$. The row associated with $s$ is therefore the function $E \to \{0, 1\}$ given by $e \mapsto \mathcal{M}(se)$, which represents the *residual language* after reading $s$.

Whenever the table is *closed* (every row for $s \in S\Sigma$ coincides with some row for $S$) and *consistent* (rows that agree on $E$ have successors that also agree), $L^\star$ constructs a hypothesis DFA whose states correspond to distinct rows in $S$. This hypothesis is submitted as an EQ. If the teacher returns a counterexample word $w$, the algorithm adds suitable prefixes of $w$ to $S$ or suffixes to $E$, refines the table, and repeats. Under an exact MAT, $L^\star$ converges to the unique minimal DFA for $\mathcal{M}$ using a number of MQs and EQs polynomial in the number of states of $\mathcal{M}$.

## 3 PROBABILISTIC MATS

### 3.1 THE pMAT FORMULATION

We generalize Angluin's Minimally Adequate Teacher (MAT) model to a setting in which answers to MQs may be *probabilistically* wrong while answers to EQs remain correct. Let the (unknown) target DFA be $\mathcal{M} : \Sigma^* \to \{0, 1\}$. A learner $\mathcal{L}$ interacts with two oracles: an (imperfect) membership oracle $O_{MQ}$ and a (perfect) equivalence oracle $O_{EQ}$.

**Definition 3.1** (pMAT). *Given a target DFA $\mathcal{M}$ over $\Sigma$, a probabilistic MAT (pMAT) is a pair $(O_{MQ}, O_{EQ})$ with the following behavior.*

- ***Membership oracle $O_{MQ}$ (stochastic).*** *On input a membership-query instance $\pi$, the oracle returns $X \sim P(\cdot \mid \pi)$ on $\{0, 1\}$.*

  *Here $\pi$ is the complete LLM-facing query object: it may include the system prompt, target-language definition, alphabet, examples, queried word, and any auxiliary labels or proposed answers included in the prompt. Let $w(\pi) \in \Sigma^*$ denote the word whose membership label is being queried. Repeating the same $\pi$ yields i.i.d. draws from $P(\cdot \mid \pi)$.*

- ***Equivalence oracle $O_{EQ}$ (exact).*** *On input a hypothesis DFA $\widehat{\mathcal{M}}$, $O_{EQ}$ outputs* OK *if $\widehat{\mathcal{M}} \equiv \mathcal{M}$, else a noise-free counterexample word $w$ with $\widehat{\mathcal{M}}(w) \neq \mathcal{M}(w)$.*

**Running example.** Figure 1 depicts a typical pMAT interaction on the parity-of-$a$ language over $\Sigma = \{a, b\}$. The learner queries the stochastic membership oracle, e.g., "Is aab in the target language?" and receives a (possibly noisy) MQ answer (*Yes* in the illustration). When the current hypothesis is submitted to $O_{EQ}$, the perfect equivalence oracle may return a CE such as ab|FALSE. Incorporating this counterexample updates the hypothesis from an incorrect two-state automaton (left) to the correct parity DFA (right). The figure emphasizes the two roles: $O_{MQ}$ is informative yet fallible; $O_{EQ}$ is the noise-free verifier that prevents drift due to systematic MQ mistakes.

For a single membership-query instance $\pi$, define the unknown correctness probability

$$p^\star(\pi) = \Pr_{X \sim P(\cdot \mid \pi)}[X = \mathcal{M}(w(\pi))], \qquad \epsilon(\pi) = 1 - p^\star(\pi).$$

When a fixed context $c$ is instantiated with a queried word $w$, write $\pi_c(w)$ for the resulting MQ instance. Then $p^\star(c, w) := p^\star(\pi_c(w)), \epsilon(w \mid c) := \epsilon(\pi_c(w))$.

Let $n^\star$ be the number of MQs used by an optimal exact-MAT learner (e.g., $L^\star$) when $O_{EQ}$ returns shortest counterexamples. The average MQ mistake rate along such a minimal set $\{w_1, \ldots, w_{n^\star}\}$ is

$$\mathcal{E}(c) = \frac{1}{n^\star} \sum_{i=1}^{n^\star} \epsilon(w_i \mid c).$$

In pMAT, these mistakes arise from approximate MQ oracles and—critically—*persist* across resampling unless corrected by external verification.

Fix a membership-query instance $\pi$. Consider i.i.d. samples $X_1, \ldots, X_m \sim P(\cdot \mid \pi)$ and any resampling-based aggregator $A_m(X_{1:m}) \in \{0, 1\}$ (majority vote, self-consistency, etc.).

**Definition 3.2** (Persistent error). *A membership-query instance $\pi$ exhibits a persistent error if the mode of the oracle's distribution disagrees with the true label of the queried word:*

$$\operatorname*{mode}_{x \in \{0,1\}} P(x \mid \pi) \neq \mathcal{M}(w(\pi)) \quad \Longleftrightarrow \quad p^\star(\pi) < \tfrac{1}{2}.$$

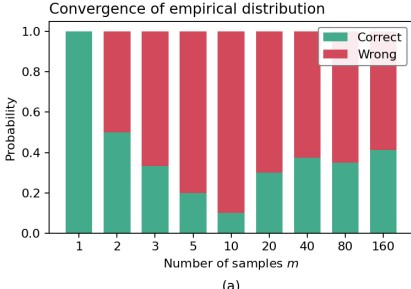 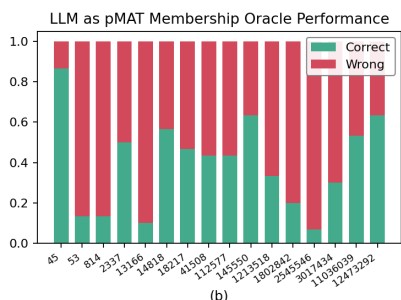

Figure 2: **Empirical behavior of LLM MQ oracles.** **(a)** Convergence of the empirical distribution under resampling for a query with $p^\star < \frac{1}{2}$: as the number of samples $m$ grows, the empirical mode concentrates on the *wrong* label (variance $\downarrow$, bias unchanged). **(b)** Per-query correctness probabilities for a set of binary strings in the divisible-by-3 language; bars show the LLM's probability of answering an MQ correctly (green) vs. incorrectly (red). Several queries have green $< 50\%$, i.e., persistent errors. Model: `GPT-4o-mini`, temperature 0.9, 30 samples per query.

## 3.2 LLMs as pMAT membership oracles

LLMs provide a concrete instantiation of the stochastic membership oracle $O_{MQ}$ in pMAT. Given a full MQ instance $\pi$, an LLM induces a conditional distribution $p_\theta(\mathbf{y} \mid \pi)$. After post-processing to a binary MQ decision, we obtain

$$O_{MQ}[\pi] \sim P(\cdot \mid \pi) = \text{Bernoulli}\big(p^\star(\pi)\big), \qquad \epsilon(\pi) = 1 - p^\star(\pi).$$

Multiple forward passes on the same $\pi$ simply draw additional i.i.d. samples from the same $P(\cdot \mid \pi)$.

**Self-consistency amplifies bias (Fig. 2a).** By the Strong Law of Large Numbers, any aggregator that asymptotically returns the empirical mode satisfies

$$A_m(X_{1:m}) \xrightarrow[m \to \infty]{\text{a.s.}} \text{mode}_x P(x \mid \pi).$$

Hence resampling reduces *variance* but cannot remove *bias*: if $p^\star(\pi) < \frac{1}{2}$, then

$$\Pr\big[A_m(X_{1:m}) \neq \mathcal{M}(w(\pi))\big] \longrightarrow 1,$$

and majority vote converges almost surely to the *wrong* label. Figure 2a visualizes this phenomenon: as the number of samples $m$ increases, the empirical distribution collapses onto a single label—but when the oracle is biased, that label is incorrect. Thus, re-sampling increases consistency while leaving systematic mistakes intact unless a noise-free verifier intervenes.

**Empirical evidence on a structured language (Fig. 2b).** We probe an LLM on the "divisible-by-3" regular language using a fixed prompt (context $c$). Each bar on the $x$-axis corresponds to a queried word $w$ (shown in decimal for legibility). The green segment indicates the probability the model labels the MQ correctly; the red segment indicates the probability of an incorrect label. Several words satisfy $p^\star(c, w) < \frac{1}{2}$ (green $< 50\%$), demonstrating *persistent errors*: resampling or majority vote will converge to the wrong answer for these MQ instances. Together with the schematic in Fig. 1, these plots justify the pMAT assumption for LLMs: MQ answers are informative yet noisy in a biased, query-dependent manner, while a perfect EQ oracle is required to detect and correct the resulting systematic mistakes.

**Prompting strategies.** Under pMAT, the context $c$ determines how a queried word $w$ is packaged into the full MQ instance $\pi_c(w)$, including the language definition, alphabet, examples, and instructions. Different prompt styles induce different distributions $P(\cdot \mid \pi_c(w))$ and thus different error profiles $\epsilon(\pi_c(w))$. We explored four prompting families: **Baseline** (direct `True`/`False` answers), **CoT** (chain-of-thought reasoning before answering), **Verification** (a second pass that checks CoT consistency), and **Discrimination** (contrastive examples drawn from a discrimination tree). Each aims to improve $p^\star(\pi_c(w))$ by encouraging consistency, coherence, or analogical reasoning. Full prompt templates are given in Appendix A.5.

---

**Algorithm 1:** Noise-Robust $L^\star$ with CE Labels and a Discrimination Tree

---

**Input:** Alphabet $\Sigma$; oracles $(O_{MQ}, O_{EQ})$; significance $\alpha$; tolerance cap $\tau_{\max}$.
**Output:** Hypothesis DFA $\mathcal{H}$.

1  $S \leftarrow \{\varepsilon\}$; $\quad E_{\text{core}} \leftarrow \emptyset$; $\quad E_{\text{pool}} \leftarrow \{\text{short suffixes}\}$.
2  Estimate $\hat{\eta}$ with a small-budget routine; set $p_0 \leftarrow 2\hat{\eta}(1 - \hat{\eta})$.
3  **while** *true* **do**
  // (1) Build classes with noise-aware same-state test
4   Use union–find on $S$ with $\text{SAMESTATE}(u, v; p_0, \alpha)$, cache negatives and early-stop.
5   Choose canonical representatives $\text{rep}(C) \in \Sigma^*$ for classes $C$.
  // (2) Closedness
6   **while** $\exists s \in S, a \in \Sigma \text{ s.t. } \nexists u \in S \text{ with } sa \sim u$ **do**
7    Insert all prefixes of $sa$ into $S$ and continue.

  // (3) Consistency via a discrimination tree
8   Build a DT $\mathcal{T}$ from $E_{\text{core}}$ and the reps $\text{rep}(C)$.
9   **if** $\exists C, a$ *whose successors lead to two distinct classes* $C_1 \neq C_2$ **then**
10   $e \leftarrow \text{LCA\_discr}_{\mathcal{T}}(C_1, C_2)$.
11   **if** $e \neq \bot$ **then** add the single column $a{+}e$ to $E_{\text{core}}$; **continue**
12   **else** enumerate short $a{+}\text{tail}$; $\text{tail} \in \Sigma^*$ and add the first separator; **continue**
  // (4) Build hypothesis $\mathcal{H}$
13  Define transitions $\delta^{\mathcal{H}}(C, a) = C'$ where $\text{rep}(C)a \sim \text{rep}(C')$.
14  For acceptance: if stored CEs reaching $C$ have a common gold label $\ell$, set $C$'s acceptance
  to $\ell$; else use majority over $\{\widetilde{y}(s) : s \in C\}$.
15  **if** $O_{EQ}[\mathcal{H}] = \text{OK}$ **then**
16   **return** $\mathcal{H}$
  // (5) Process counterexample $z \in \Sigma^*$ from $O_{EQ}$
17  *Label:* set $C_{MQ}[z] \leftarrow 1 - \mathcal{H}(z)$
18  *RS split:* find earliest $z = uae$ with $\delta^{\mathcal{H}}(\mathcal{H}[u], a) \neq \mathcal{H}[ua]$.
19  **if** *found* **then** add $e$ to $E_{\text{core}}$; insert prefixes of $z$ into $S$
20  **else** (label-only) record $z$; on repeat of the same $z$, add all suffixes of $z$ to $E_{\text{core}}$

---

**Code-based oracle.** Unlike prompt-based methods, the code-based oracle uses the LLM only once to synthesize an executable predicate (e.g., Python) from the natural-language description of the DFA. All MQs are then answered deterministically by evaluating this predicate. This eliminates variance, lowers error rates, and reduces LLM cost to a single call per task, converting free-form responses into verifiable logic. See Appendix A.5 for examples and implementation details.

## 4 CAPAL ALGORITHM

We operate in the pMAT setting (Section 3), where the learner $\mathcal{L}$ interacts with a *probabilistic* membership oracle $O_{MQ}$ and a *perfect* equivalence oracle $O_{EQ}$ to identify the unknown target DFA $\mathcal{M} = \langle Q^{\mathcal{M}}, \Sigma, q_0^{\mathcal{M}}, \delta^{\mathcal{M}}, F^{\mathcal{M}} \rangle$. Membership answers are persistently noisy: for any word $w \in \Sigma^*$, the (cached) response $C_{MQ}[w] \in \{0, 1\}$ equals the true label $\mathcal{M}(q_0^{\mathcal{M}}, w)$ with expected probability $1 - \eta$ and is flipped with expected probability $\eta < \frac{1}{2}$; asking $w$ again yields the same (persistent) value since repeated queries will not converge to the ground truth, we therefore cache the initial response. The EQ is exact: on a hypothesis DFA $\mathcal{H}$, $O_{EQ}$ returns either OK or a counterexample $z \in \Sigma^*$ with $\mathcal{H}(z) \neq \mathcal{M}(z)$.

We maintain a prefix-closed set $S \subseteq \Sigma^*$ and a multiset of suffixes $E = E_{\text{core}} \cup E_{\text{pool}} \subseteq \Sigma^*$ to construct an observation table in the $L^\star$ algorithm (Angluin, 1987). For $u \in S$ and $e \in E$, we store the cached membership label $\widetilde{y}(u \cdot e) = C_{MQ}[u \cdot e]$ like LEARNANYWAY (Angluin et al., 1997). The row of $u$ is the vector $\text{row}(u) = (\widetilde{y}(u \cdot e))_{e \in E}$. The subset $E_{\text{core}}$ contains *must-use* discriminators for state comparisons; $E_{\text{pool}}$ provides extra columns to increase statistical power but is sampled under a fixed budget.

**Noise-aware class query.** Given an estimate $\hat{\eta}$ of the membership error rate, we define the *noise floor* $p_0 = 2\hat{\eta}(1 - \hat{\eta})$. For $u, v \in S$ let

$$D(u,v) \;=\; \frac{1}{m}\,\big|\{\, e \in \mathcal{E} : \; \widetilde{y}(ue) \neq \widetilde{y}(ve) \,\}\big|, \quad \mathcal{E} := E_{\mathrm{core}} \cup E_{\mathrm{pool}}),$$

with $m = |\mathcal{E}|$. We use $\mathrm{SAMESTATE}(u, v; p_0, \alpha)$ to represent the class query: it accepts *same state* "$u \sim v$" (the noise-aware class query accepts $u$ and $v$ as belonging to the same Myhill–Nerode class) iff $D(u,v) \leq p_0 + \tau$, where $\tau = \min\{\tau_{\max}, \sqrt{\ln(2/\alpha)/(2m)}\}$ (definitions of $\alpha$ and $m$ in App. A.3.2). We cache only negative outcomes. During a comparison with $m$ planned suffixes, if the current disagreement count $d_t$ after $t$ inspected suffixes satisfies $d_t/m > p_0 + \tau$, we immediately declare and cache $(u, v) \mapsto \mathrm{DIFFERENT}$. Cached negatives are reused only for the same suffix set and threshold. Using this test, we maintain classes over $S$ via union–find. We estimate $\hat{\eta}$ with a tiny bootstrap and a reuse-first procedure that limits new MQs via a strict budget. This only affects the threshold $p_0$ and does not change correctness. We run a micro-bootstrap to estimate $\hat{\eta}$ and a confidence radius that feeds the Hoeffding tolerance. We conservatively cap $\hat{\eta} \leq \bar{\eta}$ with $\bar{\eta} \geq \eta$; this raises $p_0$ and makes false "DIFFERENT" decisions harder, which is required for the soundness of negative-result caching (details in App. A.3.2).

**Closedness and consistency** *Closedness* requires that for every $s \in S$ and $a \in \Sigma$ there exists $u \in S$ with $sa \sim u$; violations cause us to insert all prefixes of $sa$ into $S$. Closedness alone does not guarantee a DFA; we also enforce *consistency*: whenever $s_1 \sim s_2$ and $a \in \Sigma$, their successors $s_1 a$ and $s_2 a$ must land in the same class. To repair inconsistencies with few columns, we maintain a *discrimination tree* (DT) $\mathcal{T}$ (Isberner et al., 2014) over current classes: internal nodes store discriminators $e \in E_{\mathrm{core}}$; leaves store disjoint sets of classes. For two classes $C_1 \neq C_2$, the lowest common ancestor (LCA) carries a discriminator $e_{\mathrm{LCA}}$ that separates them. Thus, when a class $C$ has successors to two distinct classes under $a$, we add a *single* column $a + e_{\mathrm{LCA}}$ to $E_{\mathrm{core}}$ (and lazily cache the needed cells). If the LCA discriminator fails to separate due to noise on a representative, we fall back to a short enumerated search for a short $a + e$ that separates. By ensuring the closedness and consistency of the observation table, we can theoretically guarantee the correctness of the learned DFA. We put the proof in the Appendix A.3.

**Hypothesis construction and refinement** For each class $C$, pick a canonical representative $\mathrm{rep}(C)$ (e.g., short-lex). Define transitions by $\delta^{\mathcal{H}}(C, a) = C'$ such that $\mathrm{rep}(C)a \sim \mathrm{rep}(C')$. For acceptance, if stored counterexamples reaching a class $C$ have a common gold label $\ell$, we set $C$'s acceptance to $\ell$; otherwise, if no such gold label is available, we decide by majority over $\{\widetilde{y}(s) : s \in C\}$. If two stored counterexamples with different gold labels reach the same current class, we treat this as evidence that the current suffix set is insufficient and trigger a label-only refinement. When $O_{EQ}$ returns $z \in \Sigma^*$ with $\mathcal{H}(z) \neq \mathcal{M}(z)$, we attempt RS analysis: scan $z = uae$ for the earliest position at which

$$\delta^{\mathcal{H}}(\mathcal{H}[u], a) \neq \mathcal{H}[ua].$$

If found, we add $e$ to $E_{\mathrm{core}}$ and insert prefixes of $z$ into $S$. If no split is found, then $z$ is label-only: we record the CE label and do not add columns the first time; only if the same $z$ reappears do we minimally escalate by adding all suffixes of $z$ to $E_{\mathrm{core}}$.

This is purely for MQ efficiency: when an RS split is absent, the CE is likely due to a state mislabel rather than a structural error. If the *same* label-only CE reappears, this suggests that the current suffix set cannot isolate the erroneous class, which then justifies adding all suffixes of $z$ to $E_{\mathrm{core}}$.

The algorithm combines $L^\star$'s table, a noise-aware statistical equality test with negative caching, CE label overrides for states, and a TTT-style DT for near-optimal consistency repairs. It converges under persistent MQ noise with an exact EQ, while keeping both the number of discriminators and the number of MQs small.

**Theorem 1** (Query complexity). *Let the target minimal DFA have (n) states, alphabet $\Sigma$, access diameter $\rho$, and counterexamples of length at most $L_{\mathrm{CE}}$. Under a perfect EQ and persistent MQ noise with $\eta < \frac{1}{2}$, CAPAL terminates w.h.p. and returns the exact language. Moreover,*

$$\#EQ = O(n), \quad \#MQ = O(|\Sigma|, n^2\rho + nL_{\mathrm{CE}}) + O(m, |\Sigma|, n\rho),$$

*where $m = O(\gamma^{-2}\log(1/\alpha))$ is the suffix budget per class decision for noise margin $\gamma$ at confidence $1 - \alpha$. For fixed noise/confidence, (m) is a constant and the asymptotics match classical $L^\star$. (Proof*

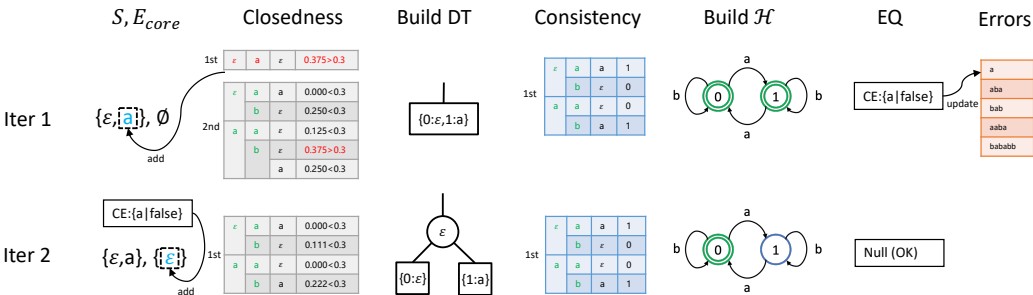

Figure 3: **Running example for CAPAL.** One CAPAL run on the DFA SIMPLE-01 (even number of $a$'s); each row is an outer iteration. The *Closedness* block enumerates all $(s, a)$ with $s \in S$ and $a \in \Sigma$; red cells mark pairs violating $\exists u \in S : sa \sim u$. The remaining blocks show, respectively, the discrimination tree (internal nodes are discriminators, leaves are classes written as {ID: MEM-BERS}), the consistency table (class IDs reached by $sa$), the current hypothesis DFA (double green circles are accepting states), the EQ result, and the set of persistent MQ errors.

*in App. A.3) DT bounds $|E_{core}| \leq n - 1$; closedness/consistency touch $O(|\Sigma|, |S|)$ pairs with $|S| = O(n\rho)$; each class decision uses at most $m$ suffixes; counterexamples contribute $O(nL_{CE})$.*

**Running Example** Figure 3 shows one CAPAL run on the DFA SIMPLE-01, which accepts exactly the words over $\Sigma = \{a, b\}$ with an even number of $a$'s. We connect CAPAL to a pMAT with persistent membership noise $\eta = 0.1$ and a perfect EQ oracle. A short bootstrap makes a few dozen MQs, estimates $\hat{\eta}$, and thus fixes the noise floor $p_0$ and threshold $p_0 + \tau$ that appear in the rightmost column of the **Closedness** tables. Words whose MQ labels are flipped by the noisy oracle are listed in the *Errors* column.

**Iteration 1.** We start with the prefix set $S = \{\varepsilon\}$ and empty core suffix set $E_{core} = \emptyset$. The first closedness check examines $s = \varepsilon$, $a = a$ and fails because there is no $u \in S$ with $sa = a \sim u$. The offending word $a$ (and its prefixes) is therefore added, giving $S = \{\varepsilon, a\}$, and a second closedness pass over this $S$ and $\Sigma$ then succeeds for all pairs $(s, a)$. Union–find on $S$ yields two classes with representatives $\varepsilon$ and $a$; since $E_{core}$ is still empty, the discrimination tree collapses to a single leaf and the consistency check is trivially satisfied (each class has one member). CAPAL builds a two-state hypothesis $\mathcal{H}_1$ whose transitions already match the even/odd-$a$ DFA, but because $a$ has been persistently mis-labeled, both states are marked accepting, so the EQ oracle returns the counterexample $c = a$.

**Iteration 2.** Using the trusted EQ label, CAPAL first corrects the label of $a$. The Rivest–Schapire scan finds no transition mismatch along $c$, so $c$ is treated as a label-only counterexample and its short suffix $a$ is promoted into $E_{core}$, while $S$ remains $\{\varepsilon, a\}$ and closedness still holds. Rebuilding the discrimination tree with core suffix $a$ now separates the two classes: appending $a$ to $\varepsilon$ and to $a$ leads to different acceptance, so they fall into distinct leaves. The **Consistency** table confirms that for every class $C$ and symbol $a \in \Sigma$, all $s \in C$ have successors $sa$ in a single class, so no further refinement is needed. Rebuilding the hypothesis $\mathcal{H}_2$ keeps the same transition graph but now only the even-$a$ state is accepting; the EQ oracle finds no further counterexample, and CAPAL halts with the correct even-parity DFA.

## 5 EMPIRICAL RESULTS

This section evaluates our prompting strategies and the CAPAL algorithm, comparing them against baseline active/passive learners.

**Setup.** Implementations are from LearnLib Isberner et al. (2015). We use 200 human–interpretable DFAs (2–50 states) from Sipser (1996); RegexLib.com (2025); Kushman & Barzilay (2013). Each DFA can be described by a short natural-language specification; the LLM oracle receives this description and the alphabet. We vary counterexample length and the pMAT

Table 1: **LLM query cost and oracle error.** Average $\pm$ std. LLM calls per DFA for LEARNANY-WAY (LA) and CAPAL; $\epsilon$ is the empirical MQ error rate. We omit TTT and $L^*$ here—when they fail to learn the harder DFAs, their query counts are meaningless for comparison.

| Oracle Type | LA | CAPAL | $\epsilon$ |
|---|---|---|---|
| Baseline | 46.07 ($\pm$33.0) | 43.75 ($\pm$30.2) | 0.156 |
| CoT | 52.64 ($\pm$37.3) | 30.93 ($\pm$16.5) | 0.068 |
| Verification & CoT | 51.57 ($\pm$36.5) | 28.96 ($\pm$19.7) | 0.044 |
| Discriminator & CoT | 51.82 ($\pm$40.1) | 29.14 ($\pm$15.8) | 0.047 |
| **Code-based** | **1** ($\pm$0) | **1** ($\pm$0) | **0.013** |

Table 2: **Learning accuracy** (fraction of DFAs learned exactly). Here TTT and $L^*$ are included because accuracy remains comparable even when some DFAs are not learned.

| Oracle Type | TTT | $L^*$ | LA | CAPAL |
|---|---|---|---|---|
| Target DFA (upper bound) | 1.000 | 1.000 | 1.000 | 1.000 |
| Baseline | 0.107 | 0.071 | 0.107 | 0.250 |
| CoT | 0.429 | 0.393 | 0.500 | 0.571 |
| Verification & CoT | 0.464 | 0.429 | 0.714 | 0.679 |
| Discriminator & CoT | 0.571 | 0.536 | 0.714 | 0.821 |
| **Code-based** | 0.750 | 0.678 | 0.786 | **0.928** |

membership error rate $\epsilon$ to probe robustness. Unless stated otherwise, the oracle is `gpt-4o` with temperature 0.9.

## 5.1 PROMPT STUDY

Table 1 compares different prompting methods. Moving from a direct **Baseline** prompt to **CoT** more than halves the MQ error rate ($\epsilon$: $0.156 \rightarrow 0.068$, $-56\%$). Adding an explicit **Verification** step or our **Discriminator**-guided hints drives $\epsilon$ to $\approx 0.045$ (0.044 and 0.047, $-72\%$ and $-70\%$ vs. Baseline). The query budgets mirror this improvement when the same learner reuses consistent answers: for example, CAPAL's LLM calls drop from 43.75 (Baseline) to 30.93 (CoT, $-29\%$), and further to 28.96 (Verification, $-34\%$) or 29.14 (Discriminator, $-33\%$). In contrast, richer prompts that perform explicit reasoning/verification can add a small constant overhead for learners that query the LLM on every MQ (see the LA column), even though their $\epsilon$ is lower.

The **code-based oracle** changes the regime entirely: a single synthesis call produces a deterministic checker, after which MQs are answered without further LLM interaction. This yields the lowest error rate ($\epsilon = 0.013$, $-92\%$ vs. Baseline) and collapses the LLM budget to **1** call per task for *both* learners ($\geq 97\%$ reduction relative to any prompting-only method).

## 5.2 ALGORITHM COMPARISON

Table 2 summarizes end-to-end success. Standard active learners (TTT, $L^*$) are brittle under noisy MQs, rarely recovering from systematic mislabels. LEARNANYWAY (LA) improves robustness via counterexamples, but CAPAL is consistently as good or better across oracle types; the gap is largest when prompts reduce $\epsilon$ (e.g., **CAPAL** 0.821 vs. LA 0.714 with Discriminator&CoT; 0.928 vs. 0.786 with **code-based**).

Then we discuss how class queries and discrimination trees of CAPAL helps:

**Class queries tame persistent noise.** Instead of trusting single MQ labels, CAPAL asks whether two prefixes $u, v$ belong to the same Myhill–Nerode class. This is decided by aggregating MQ evidence over a small set of suffixes. Under pMAT, if each MQ has correctness probability $p^\star(c, \cdot) > \frac{1}{2}$, then a majority on a handful of suffixes yields a *statistically reliable* class decision, even when any *individual* MQ may be wrong. Thus, CAPAL converts noisy Bernoulli answers into a robust hypothesis-refinement signal that is resilient to persistent, query-dependent biases.

Figure 4: LEARNANYWAY, RPNI-EDSM, CAPAL comparison as $\epsilon$ varies. Both RPNI-EDSM and CAPAL learn within $5 \times 10^4$ MQs; CAPAL requires fewer MQs than LA and keeps counterexample use stable.

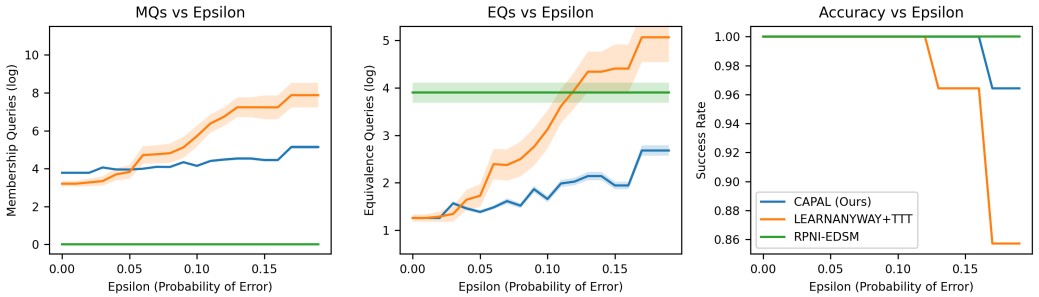

**Discrimination trees localize and minimize queries.** Rather than maintaining a large observation table (as in $L^*$) or repeatedly repairing with long counterexamples (as in LA), a discrimination tree indexes states by short distinguishing suffixes. New information only traverses a single root-to-leaf path, so each refinement reuses previously validated separators and touches $O(\log |Q|)$ tests. This (i) avoids re-asking brittle MQs; (ii) amortizes errors across repeated tests on the same separators; and (iii) naturally favors near-minimal discriminators, reducing both query count and exposure to error on long words.

## 5.3 SCALING WITH PERSISTENT NOISE

To stress-test robustness, we inject a probabilistic oracle with error rate $\epsilon \in [0, 0.2]$ and allow counterexamples up to length 10. Figure 4 shows that CAPAL uses fewer MQs than LA as noise grows and does *not* require polynomially more counterexamples. When $\epsilon \gtrsim 0.18$, purely passive RPNI-EDSM can be more query-efficient because it avoids MQs altogether; however, the prompt strategies in Table 1 typically keep $\epsilon < 0.1$, where CAPAL clearly outperforms both LA and RPNI-EDSM.

## 6 DISCUSSION AND CONCLUSION

CAPAL demonstrates that active learning of regular languages can be made robust to *persistent* membership noise when a perfect equivalence oracle is available. The central shift is from trusting individual MQ labels to issuing a *class query* that aggregates evidence against a principled noise floor, with early stopping and monotone (negative) caching to avoid re-litigating pairs already shown different. On top of the observation-table loop, a discrimination tree selects a *single* derived column $a + e_{\text{LCA}}$ to repair each inconsistency, which keeps $|E_{\text{core}}|$ linear in the number of target states. Counterexamples supply bias-free labels; we use them to override acceptance for the reached hypothesis state and to gate processing of label-only CEs, thereby preventing bursts of fresh MQs that would inject additional persistent errors.

Limitations suggest directions for future work. If $O_{EQ}$ is approximate, false negatives (missed CEs) require extending the theory to tolerate occasional incorrect OK responses. If $\hat{\eta}$ underestimates the true error rate, negative caching could in principle lock in spurious splits; our conservative cap mitigates this, but a fully adaptive, PAC-style control across many tests is desirable. Finally, persistent noise may drift with prompt context; adaptive re-calibration and drift detection would strengthen robustness.

Beyond DFAs, the CAPAL pattern—class queries, DT-guided repairs, CE-label overrides, and gated CE handling—extends naturally to Mealy/Moore machines, register automata (with domain-aware discriminators), and symbolic automata. Overall, pMAT plus CAPAL offers a practical route to *verifiable* neuro-symbolic systems: even when language models are fallible, structured interaction with a perfect equivalence oracle suffices to synthesize formal artifacts whose correctness can be certified at the end of the loop.

**Reproducibility Statement.** We have taken several steps to ensure the reproducibility of our results. All code and datasets used in this work are released on Github at `https://github.com/lkwargs/CAPAL`. Detailed descriptions of experimental settings (e.g., random seeds, hyperparameters, query budgets, and noise models) are provided in Section 5 and the Appendix. Prompt templates for all oracle variants are also included in Appendix A.5. Together, these materials enable independent researchers to reproduce all experiments and results reported in this paper.

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

## A APPENDIX

### A.1 RELATIONSHIP BETWEEN ERRORS AND CE LENGTH

Figure 5: Relationship between errors and CE length. The average length of CEs refers to the length parameter used during generating CEs. They must be longer than this length parameter unless no longer counterexample exists. If the average CE length is set to 0, the oracle returns the shortest counterexample. This test is running on a simple automaton that recognizes strings with more than 3 'a's and more than 2 'b's, employing the LEARNANYWAY & TTT as the DFA learner.

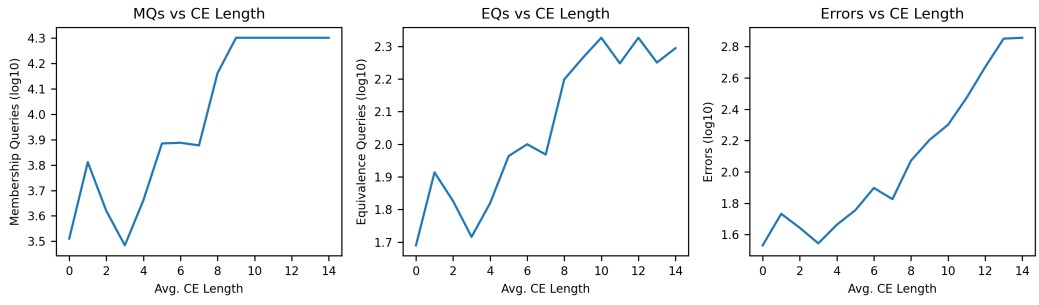

### A.2 EFFECTS AND SOLUTION OF AN IMPERFECT EQ ORACLE

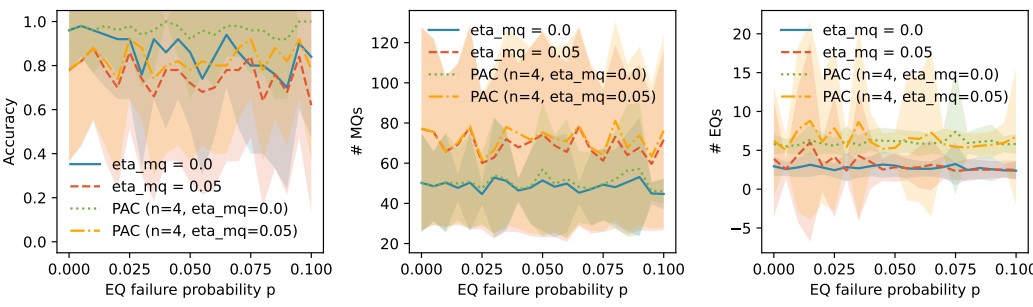

Figure 6: **Robustness to imperfect equivalence oracle.** We corrupt the EQ oracle so that whenever the hypothesis is *not* equivalent to the target, it independently fails to return a counterexample with probability $p \in [0, 0.1]$ (but still answers "equivalent" correctly when the languages match). We report mean and standard deviation over repeated runs for (left) learning accuracy, (middle) number of MQs, and (right) number of EQs, under two MQ noise levels ($\eta_{MQ} = 0$ and $0.05$) and with/without a PAC–style EQ wrapper with $n = 4$ retries.

Figure 6 shows that CAPAL remains surprisingly robust even when the EQ oracle is imperfect. With a single noisy EQ call per round (`eta_mq = 0.0/0.05` curves), the learner maintains high success probability across the entire range $p \le 0.1$: accuracy typically stays in the $80$–$95\%$ range, and failures are mostly due to rare runs where the EQ oracle repeatedly misses the same counterexample and the algorithm stops early. Importantly, the average number of MQs and EQs is essentially flat in $p$, indicating that CAPAL does not become unstable or enter long oscillations even when the EQ oracle occasionally misses counterexamples.

To further harden the algorithm, we wrap each EQ query in a simple PAC-style repetition scheme: instead of calling the imperfect oracle once, CAPAL issues up to $n = 4$ independent EQ calls and only accepts equivalence if they all return "no counterexample." This reduces the probability of silently accepting an incorrect hypothesis from $p$ to at most $p^4$ (e.g., $10^{-4}$ when $p = 0.1$), while increasing the EQ budget by at most a constant factor of $n$. The dotted `PAC` curves in Fig. 6 confirm this behavior: accuracy is driven back close to $100\%$ for all tested $p$ and both MQ-noise settings, yet the number of EQs only increases modestly and the number of MQs is virtually unchanged.

Thus, even when the practical EQ oracle is approximate, CAPAL can be made highly reliable by a lightweight PAC-wrap that introduces only a small, constant-factor overhead in EQ usage.

## A.3 Correctness and Convergence in the pMAT Setting

We analyze Algorithm 1 in the pMAT model (Section 3): the learner interacts with a *perfect* equivalence oracle $O_{EQ}$ and a *persistent-noisy* membership oracle $O_{MQ}$ with error rate $\eta < \frac{1}{2}$. The target DFA is $\mathcal{M} = \langle Q^{\mathcal{M}}, \Sigma, q_0^{\mathcal{M}}, \delta^{\mathcal{M}}, F^{\mathcal{M}} \rangle$.

**What have been proved in $L^\star$/RS/TTT.** The following facts are classical and we cite them without reproving:

1. **Closed & consistent tables define a DFA hypothesis.** In the exact-MQ setting, a closed and consistent observation table induces a well-defined DFA $\mathcal{H}$ whose states correspond to distinct rows and whose transitions are induced by row successors (Angluin, 1987; Angluin et al., 1997).

2. **Counterexamples enforce progress via suffixes.** Rivest–Schapire decomposition extracts, from any CE $z$, a short suffix $e$ that distinguishes appropriate rows; adding $e$ strictly increases the table's discriminating power (Rivest & Schapire, 1989).

3. **Discrimination trees are $O(|Q^{\mathcal{M}}|)$.** A TTT-style discrimination tree (DT) over $n$ states uses at most $n-1$ internal discriminators and repairs inconsistencies using a *single* LCA discriminator (Isberner et al., 2014).

4. **Termination in the exact-MQ case.** With exact MQ and perfect EQ, $L^\star$ terminates and returns a DFA equivalent to the target (Angluin, 1987; Angluin et al., 1997).

**What remains to prove in our setting.** Because $O_{MQ}$ is *noisy and persistent*, while $O_{EQ}$ is perfect, we must justify:

- a noise-aware *same-state* test with negative-result caching,
- the use of *CE labels* (no MQ) to fix state acceptance,
- *gated* processing of label-only CEs to avoid noise amplification, and
- finite convergence with high probability under a safe noise bound.

### A.3.1 Soundness

**Theorem 2** (Soundness)**.** *If Algorithm 1 halts with $O_{EQ}[\mathcal{H}] = \mathsf{OK}$, then $\mathcal{L}(\mathcal{H}) = \mathcal{L}(\mathcal{M})$.*

*Proof.* Immediate from the correctness of a perfect equivalence oracle: $\mathsf{OK}$ iff the languages are equal. □

### A.3.2 Noise-aware row equality and safe negative caching

Let $\hat{\eta}$ be the learner's estimate of $\eta$, and define the noise floor $p_0 = 2\hat{\eta}(1 - \hat{\eta})$. We fix a *per-comparison significance level* $\alpha \in (0, 1)$: $\alpha$ upper-bounds the probability that a single SameState call incorrectly declares two truly equivalent rows as *different*.

For $u, v \in S$, the test SameState$(u, v; p_0, \alpha)$ chooses a finite set of suffixes

$$\mathcal{E} = E_{\text{core}} \cup (\text{a capped sample from } E_{\text{pool}}), \qquad m := |\mathcal{E}|,$$

where $m$ is exactly the number of suffixes (i.e., "votes") used in this particular class comparison. It then computes the empirical disagreement rate

$$D(u, v) \;=\; \frac{1}{m} \big| \{ e \in \mathcal{E} \;:\; \widetilde{y}(ue) \neq \widetilde{y}(ve) \} \big|$$

and compares it to the threshold $p_0 + \tau$, where

$$\tau = \min \Big\{ \tau_{\max}, \sqrt{\frac{\ln(2/\alpha)}{2m}} \Big\}.$$

We *cache* negative outcomes (declare $u \not\sim v$) and do *not* cache positives. The cache key includes $(u, v)$, the suffix-set version, and the threshold $p_0 + \tau$; if any of these changes, the comparison is recomputed.

We assume the test uses a *safe upper bound* $\bar{\eta} \geq \eta$ (e.g., by capping $\hat{\eta}$): then $p_0(\bar{\eta}) \geq 2\eta(1 - \eta)$, which makes false negatives *harder*, not easier. By Hoeffding, with $m$ samples the probability that a single comparison misclassifies a truly equal pair is at most $\alpha$; with at most $N_{\text{cmp}}$ comparisons over the run, a union bound yields a total failure probability $\delta \leq N_{\text{cmp}}\alpha$. Thus the negative cache is sound *with probability at least* $1 - \delta$.[2]

### A.3.3 Counterexamples under persistent noise

Given $\mathcal{H} \not\equiv \mathcal{M}$, $O_{EQ}$ returns $c \in \Sigma^*$ with $\mathcal{H}(c) \neq \mathcal{M}(c)$.

**Trusted CE labels.** Because $O_{EQ}$ is perfect, the CE's true label is: $C_{MQ}[c] \leftarrow 1 - \mathcal{H}(c) = \mathcal{M}(c)$. We store this as a *gold* override for the exact word $c$ and insert $c$ into $S$ (prefix-closure already ensured by the algorithm).

**Structural vs. label-only CEs.** We attempt Rivest–Schapire decomposition on $c$: if there exists an earliest factorization $z = uae$ with $\delta^{\mathcal{H}}(\mathcal{H}[u], a) \neq \mathcal{H}[ua]$, we add the bare suffix $e$ to $E_{\text{core}}$ (structural progress). Otherwise $c$ is *label-only* and we gate processing: record $C_{MQ}[z]$ and do not add columns on the first occurrence; on a repeat of the *same* $c$ we minimally escalate (add all suffixes of $z$).

**Acceptance override for CE-reached states.** When building $\mathcal{H}$, if any CE $z$ reaches a class $C$ (i.e., $\mathcal{H}[z] \in C$), we *override* $C$'s acceptance to match $C_{MQ}[z]$.[3]

### A.3.4 Finite convergence with high probability

Define the finite potential

$$\Phi = \underbrace{\left| \{\{p, q\} \subseteq Q^{\mathcal{M}} : \text{ not yet separated by } E_{\text{core}}\} \right|}_{\Phi_{\text{struct}}} + \underbrace{\left| \{\text{hypothesis states with wrong acceptance}\} \right|}_{\Phi_{\text{label}}}.$$

**Lemma 1** (Each CE decreases $\Phi$ w.h.p.). *Assume the negative cache uses $p_0(\bar{\eta})$ with $\bar{\eta} \geq \eta$ and per-comparison error probability at most $\alpha$. Then with probability at least $1 - \delta$ over all same-state tests (for a suitable union bound $\delta$), each CE returned by $O_{EQ}$ strictly decreases $\Phi$:*

- *If RS finds a split $z = uae$, adding $e$ strictly increases the separation power of $E_{\text{core}}$ (by Rivest & Schapire (1989)); hence $\Phi_{\text{struct}}$ decreases.*

- *Otherwise $c$ is label-only: the CE override forces the acceptance of the reached state, decreasing $\Phi_{\text{label}}$ permanently (or, under the positive-only variant, escalation ensures the class majority flips to the CE label after finitely many repeats).*

**Theorem 3** (Finite convergence w.h.p.). *Let $n = |Q^{\mathcal{M}}|$. Under a perfect $O_{EQ}$ and a persistent-noisy $O_{MQ}$ with $\eta < \frac{1}{2}$, and using a safe noise cap $\bar{\eta} \geq \eta$ in the same-state threshold, Algorithm 1 terminates after finitely many iterations with probability at least $1 - \delta$, and by Theorem 2 returns $\mathcal{H}$ with $\mathcal{L}(\mathcal{H}) = \mathcal{L}(\mathcal{M})$. Moreover, the number of* bare *discriminators added to $E_{\text{core}}$ is $O(n)$ and in fact at most $n - 1$ by the DT bound of Isberner et al. (2014).*

*Proof sketch.* By Lemma 1, each CE strictly decreases the finite potential $\Phi \leq \binom{n}{2} + n$ with probability at least $1 - \delta$. Hence only finitely many CEs occur before $O_{EQ}$ must answer OK, and soundness (Theorem 2) gives correctness at termination. The $O(n)$ discriminator bound follows from the cited DT result. $\square$

---

[2]Choosing a schedule $\alpha_t = \alpha_0/t^2$ makes the sum $\sum_t \alpha_t$ finite, yielding a fixed $\delta$ without needing an a-priori $N_{\text{cmp}}$.

[3]A symmetric override (both $C_{MQ}[c] = 1$ and 0 force the state's acceptance) gives the cleanest proof. The implementation may use a positive-only override; then repeated label-only CEs trigger bounded suffix additions that make the class majority agree with the CE, and the convergence argument below still goes through.

**Remarks.** (i) The only probabilistic step is the row-equality test; by increasing the capped column budget (or decreasing $\alpha$) we can make $\delta$ arbitrarily small. (ii) Using a safe cap $\bar\eta$ makes the negative cache conservative: we never lock in a spurious split due to underestimating the noise floor. (iii) Gating of label-only CEs prevents large bursts of fresh MQs that would introduce additional persistent errors; escalation on repeats is bounded and ensures eventual correction.

## A.4 COMPLEXITY ANALYSIS

Let $M$ be the minimal DFA for the target language, with $n := |Q_M|$ states and alphabet $\Sigma$. Let $\rho$ denote the *access diameter*, i.e., the maximum length of a shortest access string to any state, and let $L_{\mathrm{CE}}$ bound the length of counterexamples returned by the equivalence oracle.

**Structural complexity** CAPAL follows exactly the classic $L^\star$ loop: it maintains a prefix-closed set $S \subseteq \Sigma^\star$, a finite set of distinguishing suffixes $E_{\mathrm{core}} \subseteq \Sigma^\star$, enforces closedness and consistency, constructs a hypothesis DFA $H$, and submits it to an equivalence oracle. The only algorithmic difference is that consistency repairs are guided by a discrimination tree, which guarantees that the number of *bare* discriminators is at most $n - 1$ (TTT bound).

If access strings are kept shortest, the prefix-closed set $S$ has size

$$|S| = O(n\rho),$$

because each state admits at most one shortest access string of length at most $\rho$, and all prefixes of these strings are also included. Each time closedness and consistency are enforced, at most $|S|$ rows and $|E_{\mathrm{core}}|$ columns of the observation table are queried. With $|E_{\mathrm{core}}| \le n - 1$, this yields

$$\text{table-filling MQs} \;=\; O\big(|\Sigma|\,|S|\,|E_{\mathrm{core}}|\big) \;=\; O\big(|\Sigma|\,n^2\,\rho\big).$$

Each counterexample contributes an additional $O(L_{\mathrm{CE}})$ MQs to fill the row/column positions along the offending access path, just as in $L^\star$. Standard arguments then imply that at most $O(n)$ counterexamples are needed: each either adds one new discriminator (at most $n - 1$ in total) or fixes the acceptance of one previously misclassified state. Hence, in the noiseless case

$$\#\mathrm{EQ} = O(n), \qquad \#\mathrm{MQ} = O\big(|\Sigma|\,n^2\,\rho + nL_{\mathrm{CE}}\big),$$

which matches the usual $L^\star$ regime up to constant factors.

**Additional cost from noisy MQs and class queries** In the pMAT model, MQ answers may be incorrect with average error rate $\eta < \frac{1}{2}$ and may exhibit persistent bias on individual queries. The only place where CAPAL pays an extra cost compared to $L^\star$ is in deciding whether two prefixes $u, v \in S$ should be treated as belonging to the same Myhill–Nerode class. Instead of comparing single table rows, CAPAL uses a *class query*: it draws a multiset of suffixes $E \subseteq \Sigma^\star$, computes the empirical disagreement rate

$$D(u,v) = \frac{1}{|E|}\big|\{e \in E : y(ue) \ne y(ve)\}\big|,$$

and declares "$u \sim v$" iff $D(u,v) \le p_0 + \tau$, where $p_0 = 2\hat\eta(1 - \hat\eta)$ is a conservative noise floor and $\tau$ is a Hoeffding tolerance.

Let $\gamma(u,v) := \mathbb{E}[D(u,v)] - p_0$ denote the true gap between the disagreement rate of $(u,v)$ and the noise floor. A standard concentration argument shows that, for any fixed confidence parameter $\alpha \in (0,1)$, there exists a universal constant $c > 0$ such that

$$m := |E| \;\ge\; \frac{c}{\gamma(u,v)^2}\,\log\frac{1}{\alpha} \quad\Rightarrow\quad \Pr\Big[\text{the test misclassifies } (u,v)\Big] \;\le\; \alpha.$$

Thus each logical "same/different" decision on a pair $(u,v)$ is implemented by a class query consuming at most

$$m = O\Big(\tfrac{1}{\gamma(u,v)^2}\log\tfrac{1}{\alpha}\Big)$$

noisy MQs. In particular, when $\eta$ is bounded away from $\frac{1}{2}$ and we only require a fixed confidence level, $m$ can be treated as a constant independent of $n$, $|\Sigma|$, and $\rho$. In the noiseless limit ($\eta = 0$), one can take $m = 1$, and the class query reduces to classical row equality.

Since class queries are only needed when checking closedness and consistency, the number of logical pairs $(u, v)$ that are ever examined is at most on the order of $|\Sigma|\,|S|$ (each extension $sa$ must be related to one representative in $S$). Therefore the extra MQ cost due to class queries is

$$\text{(class-query MQs)} \;=\; O\big(m\,|\Sigma|\,|S|\big) \;=\; O\big(m\,|\Sigma|\,n\,\rho\big).$$

This term is of the same polynomial order as the table-filling term; when $m$ is treated as a constant depending only on the noise level and desired confidence, the asymptotic behavior is unchanged.

**Main complexity bound**   Combining the structural bounds with the class-query overhead and using that the number of bare discriminators is at most $n-1$ and the number of counterexamples is $O(n)$, we obtain:

**Theorem 4** (Complexity of CAPAL). *Let $M$ be a minimal DFA with $n$ states, alphabet $\Sigma$, access diameter $\rho$, and counterexample length bounded by $L_{\mathrm{CE}}$. Assume a pMAT oracle with persistent MQ noise of average rate $\eta < \frac{1}{2}$ and a perfect EQ. Then* CAPAL *terminates with high probability and returns a DFA equivalent to $M$. Moreover, for any fixed confidence parameter $\alpha$, there exists a constant $m = O\big(\frac{1}{\gamma^2} \log \frac{1}{\alpha}\big)$ depending only on the noise margin $\gamma > 0$ such that*

$$\#\mathrm{EQ} = O(n),$$

$$\#\mathrm{MQ} = O\big(|\Sigma|\,n^2\,\rho + nL_{\mathrm{CE}}\big) \;+\; O\big(m\,|\Sigma|\,n\,\rho\big).$$

*In particular, if $m$ is treated as a constant (e.g., $\eta$ bounded away from $\frac{1}{2}$ and fixed $\alpha$), the asymptotic order collapses to the classical $L^\star$ rate up to constants:*

$$\#\mathrm{EQ} = O(n), \qquad \#\mathrm{MQ} = O\big(|\Sigma|\,n^2\,\rho + nL_{\mathrm{CE}}\big).$$

*In the noiseless case ($\eta = 0$), we have $m = 1$ and* CAPAL *reduces exactly to the $L^\star$/TTT discipline.*

This matches the intended design: CAPAL preserves the query-complexity guarantees of classical active DFA learning, while paying only a controlled multiplicative overhead for the statistical class queries needed to tolerate persistent MQ noise.

## A.5   Prompt Engineering

### A.5.1   Baseline Prompt

We use a minimal instruction that states the language definition and alphabet, then asks whether a word $w \in \Sigma^*$ is in the language, requesting a terse `True`/`False` output (Fig. 7, left). This serves as our simplest MQ oracle and exposes the raw error profile $P_c(\cdot \mid w)$ under pMAT.

*Template (abridged).*

```
Language definition:  [...]
Alphabet:  [...]
Word:  [...]
Answer with exactly True or False.
```

### A.5.2   Chain-of-Thought (CoT) Prompt

To reduce shortcut errors, the CoT variant asks the model to reason before committing to a label (Fig. 7, middle/right). The final decision is still parsed as a single Boolean token, while intermediate text is ignored by the learner.

*Template (abridged).*

```
Language definition:  [...]
Alphabet:  [...]
Word:  [...]
Reason step by step.  Then output on a new line:
Final:  True or False
```

Full prompt text, examples, and ablations for both templates are provided in Figure 7.

Figure 7: An Overview of Basic Prompts

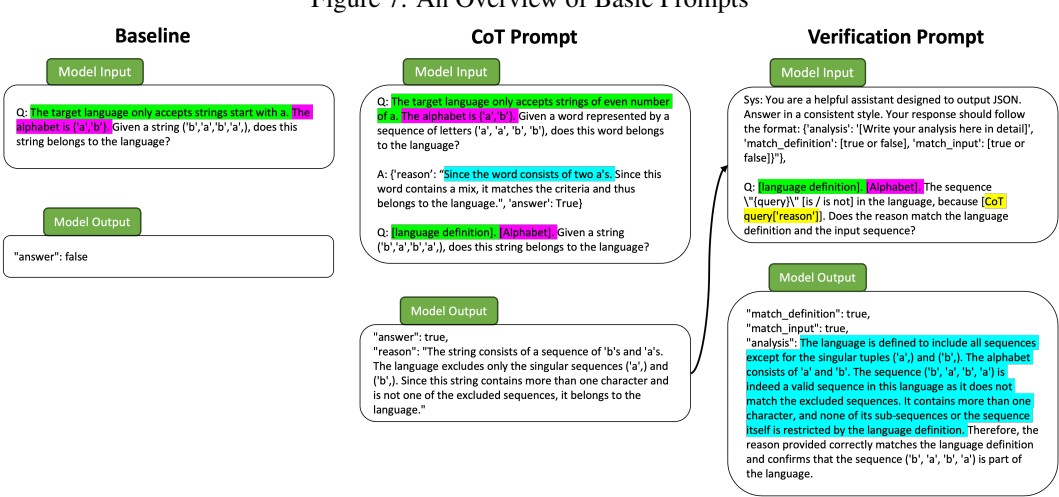

### A.5.3 VERIFICATION PROMPT

The `Verification` prompt extends the capabilities of the Chain of Thought (CoT) prompts. Initially, LLMs respond to membership queries by performing analyses similar to those demonstrated in example queries as we see in figure 7. However, due to the tendency of LLMs to generate hallucinated content, these analyses may not always be relevant to the actual query input or the language definitions, leading to incorrect membership decisions. To address this issue, we introduce a `Verification` step.

Upon receiving the initial response from the LLM to a CoT query, a secondary process begins where a "teacher" LLM checks whether the response aligns with the expected definitions and the context of the query. This involves a comparison of the initial analysis against a set of language definitions that constitute a correct response, as well as the original query input. The LLMs are required to output a JSON object containing three components:

- whether the analysis matches the language definition,
- whether the analysis is consistent with the query input, and
- the reasons why the analysis matches the definitions, as well as why the analysis is based on the query input.

If the teacher finds discrepancies between the initial answer and the expected standards, the following actions are taken: a) If the input sequence and definition match but the initial response was incorrect, the answer is inverted (changed from true to false, or vice versa). b) If there is a mismatch in the input sequence or definition, the system revises its additional prompt based on the teacher's analysis and re-evaluates the query. This verification step significantly enhances the reliability of the system's outputs. By ensuring that each membership query's response is not only generated but also rigorously checked, the LLMs minimize errors and align their outputs closely with accurate interpretations of the language rules.

### A.5.4 DISCRIMINATION PROMPT

The idea behind the `Discrimination` Prompt is that LLMs perform better with examples compared to direct inference. However, handling the large volume of membership queries necessary for learning an automaton is challenging, particularly when it is impractical to present all cached queries to LLMs. That is, we have to find the most similar ones to prompt LLMs. Intuitively, words that appear similar should share properties within the same language, but this is not universally applicable. See the example in figure 8. Consider an automaton in (a), for a membership query on the word $\langle abbbbb \rangle$, among the cached queries in (c). The $\langle bbbbb \rangle$ appears more similar due to a lower edit distance. Yet, prompting LLMs with $\langle bbbbb \rangle$ might lead them to incorrectly infer non-membership, mirroring its

Figure 8: Discrimination prompt running example: (a) target automaton that only accepts the string starting with $a$. (b) The corresponding discrimination tree to the target DFA. The leaves are states in the automaton. The inner nodes represent the discriminator that makes the states in left and right side different. (c) The edit distance between the new query and the cached queries.

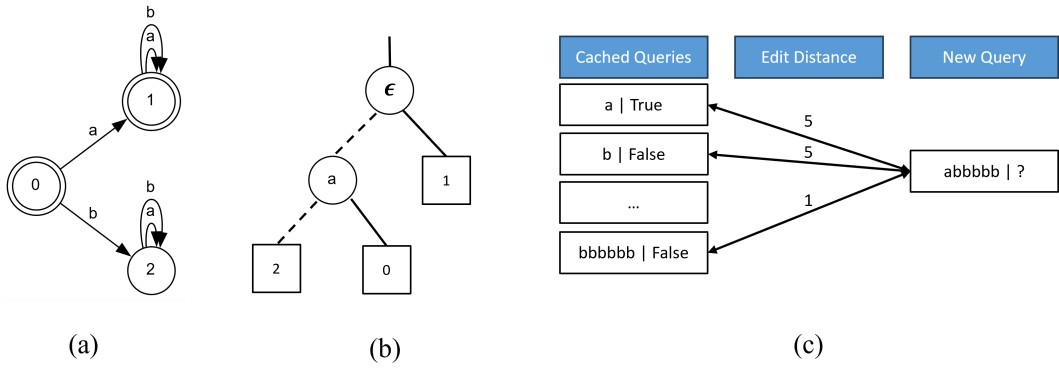

(a)                          (b)                          (c)

---

**Algorithm 2:** Find Similar Counterexamples by Discrimination Tree

---

1 **Procedure** `HistoryOracle(C, q)`:
2     **if** $q \in C_{EQ}$ **then**
3         **return** $C_{EQ}[q]$
4     **else**
5         **return** *False*

6 Represent *HistoryOracle* by $O_{\mathcal{H}}$

7 **Procedure** `DiscriminationBasedWordSearch(q)`:
8     Construct the hypothesis $\mathcal{H}$ via any active learner by $O_{\mathcal{H}}$
9     Build the discrimination tree $DT$ based on $\mathcal{H}$
10    Find the lowest common ancestor $d$ of $\mathcal{H}[q]$ and another child $s$ of $d$
11    Use Levenshtein Distance $L$ to estimate similarity between words
12    Initialize $l_q \leftarrow \infty$, $l_s \leftarrow \infty$
13    **for** *each query* $q_i \in C$ **do**
14       **if** $\mathcal{H}[q_i] = \mathcal{H}[q]$ **and** $L(q_i, q) < l_q$ **then**
15          $l_q \leftarrow L(q_i, q)$
16          $w_q \leftarrow q_i$
17       **else if** $\mathcal{H}[q_i] = \mathcal{H}[s]$ **and** $L(q_i, q) < l_s$ **then**
18          $l_s \leftarrow L(q_i, q)$
19          $w_s \leftarrow q_i$
20    **return** $w_q, w_s$

---

negative example. Conversely, using $\langle a \rangle$ as a prompt highlights that valid strings should start with 'a', though it might not instill confidence due to significant differences from the query word.

To address these challenges, we implement the `Discrimination` prompt, which maintains a discrimination tree in (b) to remember the EQs and their relationships. Upon receiving a new membership query, the algorithm identifies the query's position within the tree—specifically, which leaf (state) the new word belongs to. It then proceeds to the lowest common ancestor of that leaf and selects the most similar word by edit distance from each child of that ancestor. Words on the same leaf share the same state in the hypothesized automaton, while words on adjacent leave (sub-tree) diverge in their outputs upon receiving identical inputs. In this case, both the $\langle a \rangle$ and $\langle bbbbbb \rangle$ will be chosen to prompt LLMs. A more detailed process can be found in algorithm 2. This method

Figure 9: The Code-based Oracle

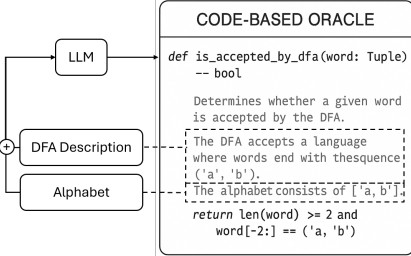

ensures that selected queries not only exhibit similarity in terms of edit distance but also align in their properties, thereby teaching the LLM what is permissible and what is not in the language.

### A.5.5 CODE-BASED ORACLE PROMPT

While current LLMs demonstrate impressive capabilities, they often struggle with consistency, leading to instability in MQ responses. Specifically, LLMs may provide contradictory answers to queries involving the same concept. However, LLMs excel at translating natural language into precise programming logic. Leveraging this strength, we propose a code-based oracle: instead of relying on direct LLM answers to MQs, we use the LLM to generate a Boolean function that formally encodes the intended concept or domain knowledge. This function acts as a deterministic existence oracle, evaluating whether a given word satisfies the desired property.

Figure 9 provides a concrete illustration of how the code-based oracle operates. Given a DFA description and its alphabet, the LLM synthesizes a Python function that determines whether a word is accepted by the target language. This function encodes the concept in executable logic, serving as a deterministic oracle. For example, if the DFA accepts words ending with the sequence (‘a’, ‘b’), the LLM generates a function that checks for this property explicitly. By converting natural language into a stable, verifiable form, the code-based oracle mitigates the inconsistency commonly observed in direct LLM responses.

To demonstrate the advantages of the code-based oracle over direct LLM querying, we compared its performance with several standard prompting approaches. Additionally, we introduced two novel prompting methods (see Appendix .c and .d) that can improve the LLM MQ answering accuracy and evaluated them alongside the code-based oracle. We compare the code-oracle with 3 prompting-based oracles to prove the necessity of code-based oracle. These comparisons are detailed in the evaluation section.

### A.6 DFA LEARNING EXAMPLES

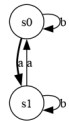

Figure 10: Simple-01: A DFA language of words containing an even number of a's. The alphabet is ['a', 'b']. The empty word is represented by ().

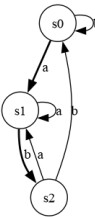

Figure 11: Simple-03: A DFA language of words that end with ('a', 'b'). The alphabet is ['a', 'b']. The empty word is represented by ().

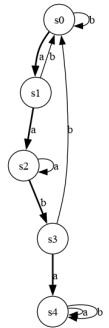

Figure 12: Simple-05: A DFA language of words containing ('a', 'a', 'b', 'a'). The alphabet is ['a', 'b']. The empty word is ().

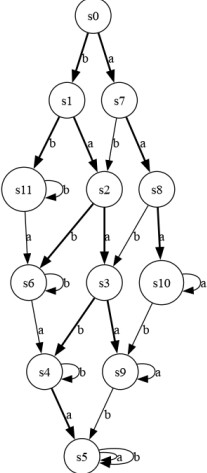

Figure 13: Normal-01: A DFA language of words with at least three a's and two b's. The alphabet is ['a', 'b']. The empty word is ().

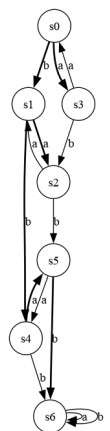

Figure 14: Normal-03: A DFA language of words with an even number of a's and one or two b's. The alphabet is ['a', 'b']. The empty word is ().

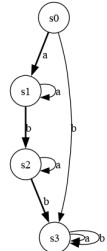

Figure 15: Normal-05: A DFA language of words starting with an a and having at most one b. The alphabet is ['a', 'b']. The empty word is ().

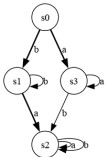

Figure 16: Difficult-03: A DFA language excluding the substrings ('a', 'b') and ('b', 'a'). The alphabet is ['a', 'b']. The empty word is ().

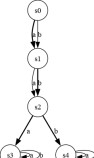

Figure 17: Difficult-09: A DFA language of words of length 3 or more, where the third symbol is an a. The alphabet is ['a', 'b']. The empty word is ().

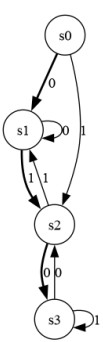

Figure 18: Difficult-12: A DFA language of binary numbers (['0', '1']) that are multiples of 3. The empty word is ().

