# OpenReview forum: "Towards Persistent Noise-Tolerant Active Learning of Regular Languages with Class Query"
_ICLR.cc/2026/Conference — ICLR 2026 Poster_

### Official Review · Reviewer_w6ef · 2025-10-29

**Soundness:** 2
**Presentation:** 1
**Contribution:** 2
**Rating:** 2
**Confidence:** 3

**Summary:**

The paper studies a new learning theory problem for regular languages, have a noisy membership query (MQ) oracle, but a perfect equivalence query oracle in a L* learning setup. Such noisy MQ oracles could be LLMs, though I am not sure why they are needed for the simple problems studied. The proposed algorithm CAPAL shows high accuracy with fewer calls to the MQ oracle.

**Strengths:**

1. I like the problem formulation of pMAT and the recognition of persistent errors.
2. The work studies a variety of prompting strategies and compares them for their effectiveness within membership query oracles.
3. The code-based oracle is pretty interesting and gives consistently better results.

**Weaknesses:**

1. My biggest criticism of the paper is in the motivation. I think the abstract and intro could provide more evidence for the utility of the problem studied from the practical perspective, with detailed, concrete examples, especially involving LLMs for membership queries. The abstract doesn't introduce the problem and is hard to follow.
2. I question whether the assumption of a perfect EQ oracle is realistic. For the experiments, how is the perfect EQ oracle implemented? How would we get a perfect EQ oracle for practical problems?
3. The main algorithm/contribution, CAPAL's description is pretty dense and hard to follow. Multiple crucial steps are introduced, but there is less background provided to fully understand and appreciate it.

I recommend rejection mainly for the paper's writing, which currently lacks motivation for the problem and its assumptions and is pretty dense. I encourage the authors to simplify the writing, maybe by including some examples, which can make their contributions more accessible.

**Questions:**

1. Why is the code based oracle deterministic? In repeated samples from the MQ oracle LLM, we could get different codes and hence even this scenario is stochastic?
2. I wonder whether the CAPAL algorithm necessarily needs LLMs. What would it look like theoretically and empirically for any other kind of  noisy MQ oracle, other than LLM?

---

> ### Author Response · Authors · 2025-11-22
> **Author Rebuttal to Reviewer w6ef**
>
> We thank the reviewer for the careful reading and constructive feedback. Below we address each concern and clarify our choices. All reference points now direct to the revised version, and all updated content is marked in **blue**.
>
> ---
>
> ### **W1**
>
> We modified the abstract and the introduction accordingly.
>
> **Motivation.**
> Automata are widely used for planning, verification, and other applications, but constructing them can be cumbersome and error-prone. A natural question is whether LLMs—whose training captures diverse, internet-scale procedural knowledge—can generate such task knowledge in automaton form. However, LLMs may hallucinate, so we must design query protocols that remain robust under probabilistic errors.
> We view this as a *renaissance of grammatical inference*, where oracles capable of answering basic queries have long been assumed but rarely available in practice. LLMs now provide a plausible instantiation of such oracles across many domains, and our method is designed to mitigate their probabilistic, persistent errors.
>
> **Practical example.**
> Consider deploying an autonomous driving agent in a new city with unfamiliar driving conventions.
>
> **LLM system prompt:**
> *“You are an autonomous vehicle driving in city X. The user will prompt you to help define steps that constitute effective driving policies in this city. When you respond, use the following variables (the automaton alphabet) to describe actions and observations.”*
>
> - **Action alphabet:** {Stop, Accelerate, Yield, turnLeft, turnRight, Honk}
> - **Pedestrian alphabet:** {closeToVehicle, waitingAtLight, distractedWhileWalking}
> - **Environment alphabet:** {redLight, greenLight, stopSign, potHole, animalCrossing, yieldSign, speedLimit, proximityToOtherVehicle}
>
> Our approach prompts the LLM to extract traces such as:
>
> - `[stopSign, Stop, Accelerate, turnRight]`
> - `[redLight, Stop, …]`
> - `[animalCrossing, Stop, …]`
> - `[potHole, Stop, …]`
>
> The first two reflect standard driving behavior; the latter two capture city-specific behaviors (frequent animals, poor infrastructure). CAPAL merges these traces into a DFA representing driving policy. An EQ oracle—here, a simulator—tests this DFA and provides counterexamples for refinement. The final DFA can be used for safety verification or as a structured reward machine to accelerate policy learning.
>
> ---
>
> ### **W2 — On assuming a perfect EQ oracle**
>
> This assumption is standard in active DFA learning (used consistently for ~40 years), and it is also **practically feasible**: in our setting the EQ oracle is *not* an LLM but a trusted verifier such as a reference DFA, simulator, or reward-machine executor, all of which can deterministically generate counterexamples.
>
> We also explicitly evaluate CAPAL under an **imperfect** EQ oracle that misses counterexamples with probability $p \le 0.1. $ CAPAL still achieves ≈80–95% success with essentially flat MQ/EQ budgets, and a simple PAC-style repetition (up to 4 EQ calls per round) restores accuracy to nearly 100% with only a small constant-factor increase in EQ usage (Appendix A.2).
>
> ---
>
> ### **W3 — CAPAL presentation is dense**
>
> We added a complete running example (the DFA for “even number of `a`”) and use it to give a step-by-step walkthrough of Lines 378–419: how `SAMESTATE` is applied, how the discrimination tree is updated, and how hypotheses are refined after each counterexample.
>
> ---
>
> ### **Q1 — “Why is the code-based oracle deterministic?”**
>
> As stated in Lines 302–306, synthesis is performed **once**: the LLM produces a pure Boolean predicate from the natural-language description. All subsequent MQs call this **fixed program**, so MQ answers are deterministic even though the *single* synthesis step is stochastic.
>
> ---
>
> ### **Q2 — “Does CAPAL need LLMs?”**
>
> No. CAPAL requires only a pMAT-style MQ oracle with *stationary* label noise bounded away from $1/2$ under fixed context. Any such oracle—human experts, heuristic checkers, fuzzers with bias, or instrumented systems with occasional mislabels—can be used.
>
> We use LLMs because:
>
> 1. They provide a compelling pMAT scenario for NL-to-formal alignment.
> 2. They empirically exhibit **persistent per-query biases** (Fig. 2b), which motivate our statistical class-query design.
>
> We will add a clarifying paragraph in Sec. 3 (pMAT is oracle-agnostic) and in Sec. 5.3 (we already stress-test with synthetic Bernoulli noise).
>
> ---
>
> We appreciate the reviewer’s pointers on motivation and exposition. We added concrete LLM-centered examples, simplified the presentation of CAPAL, and clarified both the EQ-oracle assumption and the determinism of the code-based MQ oracle. We hope these revisions address the main concerns and make the contributions more accessible.

---

### Official Review · Reviewer_6djq · 2025-10-31

**Soundness:** 3
**Presentation:** 3
**Contribution:** 3
**Rating:** 6
**Confidence:** 2

**Summary:**

This paper tackles the challenge of using Large Language Models (LLMs) to learn formal representations, specifically deterministic finite automata (DFAs).

It proposes a new learning model called the Probabilistic Minimally Adequate Teacher (pMAT). This framework assumes the LLM acts as a noisy membership query (MQ) oracle but is paired with a perfect equivalence query (EQ) oracle that provides definitive counterexamples to incorrect hypotheses.

The paper presents CAPAL (Class-query Active, Persistent-noise-Aware Learning), an algorithm designed to learn Deterministic Finite Automata (DFAs) correctly within the pMAT model. Instead of trusting individual noisy answers, CAPAL uses a statistical class query to determine if two prefixes belong to the same state and a discrimination tree to efficiently refine its hypothesis with minimal new queries.

Experiments show that CAPAL significantly outperforms standard active learning algorithms in noisy environments. The study also finds that having the LLM generate a code-based oracle (a deterministic program to answer queries) is far more effective than direct prompting, reducing errors by over 90% and cutting LLM calls to just one per task.

**Strengths:**

1.  It introduces a novel learning framework (pMAT) and a provably correct algorithm (CAPAL) to solve it. The paper formalizes the problem of "persistent noise," where an LLM is consistently wrong about certain queries, which is a more challenging and practical scenario than random errors. The proposed CAPAL algorithm is specifically designed to be robust to this noise by using a statistical "class query" and is backed by a formal proof of convergence, demonstrating its theoretical soundness.
2.  It provides a solution with the "code-based oracle." The paper's practical finding is that using an LLM a single time to synthesize a deterministic program (a code-based oracle) is far more effective than using it for repeated queries.
3. The claims are supported by a thorough and rigorous experimental evaluation. The authors validate it extensively against multiple different active and passive learning baselines across multiple benchmarks.

**Weaknesses:**

1. The entire theoretical guarantee and the practical success of CAPAL hinge on the assumption of a perfect, noise-free Equivalence Query (EQ) oracle that always returns a valid counterexample. This is a limitation, as real-world verifiers (whether human or automated) are often fallible. The paper acknowledges this but does not investigate the consequences. The work could be strengthened by including experiments with an imperfect EQ oracle (e.g., one that fails to find a counterexample with some probability).
2. The algorithm's strategy for a "label-only" counterexample (where a state is simply mislabeled as accepting/rejecting) is to wait for the exact same counterexample to reappear before escalating by adding its suffixes as discriminators . This seems potentially inefficient. A different counterexample that reaches the same mislabeled state would not trigger this escalation, possibly leading to many wasted EQ cycles to fix one incorrect label. The paper would be more convincing if it justified this specific design choice with evidence.

**Questions:**

1. The paper's entire theoretical and practical framework relies on a perfect, noise-free $O_{EQ}$. How do you expect CAPAL to perform if this oracle is imperfect? For instance, what happens if the $O_{EQ}$ fails to return a counterexample for an incorrect hypothesis (a "false negative") or provides a spurious counterexample for a correct one (a "false positive")?
2. The paper states that for label-only counterexamples (where the DFA's structure is correct but a state's acceptance is wrong), the algorithm only escalates by adding all suffixes of c to $E_{core}$ if the exact same counterexample c reappears. What is the specific rationale for this "repeat-only" escalation policy?

---

> ### Author Response · Authors · 2025-11-22
> **Rebuttal to Reviewer 6djq**
>
> We thank the reviewer for the careful reading and positive assessment. The review highlights that (i) using LLMs to guide automata learning is an interesting direction, (ii) CAPAL provides evidence that active DFA learning can be made robust to *persistent* MQ noise when a perfect EQ oracle is available, and (iii) shifting from trusting single MQ labels to class queries with early stopping and negative caching is a meaningful way to control LLM-induced noise. We are grateful for these comments and address the main concern below. All reference points now direct to the revised version, and all updated content is marked in **blue**.
>
> ---
>
> ### **W1 and Q1**
>
> Our analysis follows the standard MAT setting in active DFA learning (Angluin’s L\*, Rivest–Schapire, TTT, etc.), which has long assumed a perfect EQ oracle in order to obtain exact identification guarantees. In our pMAT instantiation this assumption is also practically feasible: the EQ oracle is *not* an LLM but a task-specific verifier (reference DFA, simulator, conformance tester, regex/policy engine), and in all experiments we implement it by constructing the difference automaton and returning a shortest separating word.
>
> To directly probe imperfect verifiers, we add experiments (Appendix A.2) where each EQ call independently fails to find a counterexample with probability $p \in [0, 0.1]$. CAPAL remains robust (success typically $80$–$95$% with essentially flat MQ/EQ budgets), and a simple PAC-style repetition of up to four EQ calls per round pushes accuracy back near $100$% with only a modest constant-factor increase in EQs.
>
> ---
>
> ### **W2 and Q2**
>
> We clarify that CAPAL does *not* wait for a repeated counterexample to fix a mislabeled state: as soon as $O_{EQ}$ returns $c$, we store its true label, insert $c$ and its prefixes into $S$, and the next hypothesis uses this gold label at the class level via the class-query mechanism.
>
> For any counterexample $c$ we then run Rivest–Schapire analysis on $c = u a e$:
>
> - If we find a structural split, we add $e$ to $E_{\mathrm{core}}$ and the prefixes of $c$ to $S$.
> - If no split is found (a pure label correction), we cache the label and deliberately skip adding columns on this first occurrence.
>
> We only escalate by adding all suffixes of $c$ to $E_{\mathrm{core}}$ if the *same* label-only counterexample reappears, thereby saving unnecessary MQs: the absence of an RS split may be caused by a simple state mislabel, whereas a repeated label-only counterexample indicates that the current suffix set cannot detect the structural error and must be expanded with all suffixes to split the state.

---

### Official Review · Reviewer_dPdj · 2025-11-01

**Soundness:** 3
**Presentation:** 3
**Contribution:** 2
**Rating:** 6
**Confidence:** 3

**Summary:**

The paper introduces a probabilistic framework for learning formal languages from large language models (LLMs) that act as noisy oracles. The authors formalize this setting through probabilistic Minimally Adequate Teachers (pMATs), which can provide incorrect answers to membership queries but always return valid counterexamples for failed equivalence queries. Within this framework, they propose CAPAL (Class-query Active, Persistent-noise-Aware Learning), an algorithm for actively learning deterministic finite automata (DFAs) that remains theoretically correct even when the oracle produces persistent labeling errors. CAPAL extends Angluin’s classic L* algorithm with statistical same-state tests that distinguish language classes under bounded noise, and with a discrimination tree intended to keep the hypothesis compact. The method also estimates the oracle’s noise rate using a lightweight bootstrap procedure that minimizes redundant queries. The authors prove convergence of CAPAL under a perfect equivalence oracle and show empirically that in certain settings the method requires fewer membership queries than existing approaches. More specifically, when applied to datasets such as RegexLib and KB13 the method shows improved robustness in DFA learning from noisy oracles. Overall, the results support the idea that LLMs, despite their fallibility, can serve as useful collaborators for synthesizing verifiable formal models from natural language.

**Strengths:**

The idea of using LLMs to guide automata learning algorithms is interesting. Their implementation CAPAL provide evidence that  active learning of regular languages can be made robust to persistent membership noise when a perfect equivalence oracle is available. The main approach is to shift from trusting individual MQ labels to issuing a class query that aggregates evidence against a principled noise floor, with early stopping and monotone (negative) caching to avoid re-litigating pairs already shown different.

**Weaknesses:**

The main problem is that the approach does not work if the equivalence oracle is approximate, and no extension to the approximate case is provided . This is an issue because perfect equivalence oracles are typically expensive, and in practice approximate equivalence oracles are used.

**Questions:**

No question.

---

> ### Author Response · Authors · 2025-11-22
> **Rebuttal to Reviewer dPdj**
>
> We thank the reviewer for the constructive comments and focus here on the scope of the equivalence-oracle (EQ) assumption and how to relax it. Below we address this concern and integrate the explanation into the revised paper. All reference points now direct to the revised version, and all updated content is marked in blue.
>
> ---
>
> ### **W1**
>
> Our analysis follows the standard MAT setting in active DFA learning (Angluin’s L\*, Rivest–Schapire, TTT, etc.), which has long assumed a **perfect EQ oracle** in order to obtain exact identification guarantees. In our pMAT instantiation this assumption is also **practically feasible**: the EQ oracle is *not* an LLM but a task-specific verifier (reference DFA, simulator, conformance tester, regex/policy engine). In all experiments we implement this oracle by constructing the difference automaton and returning a shortest separating word.
>
> To directly evaluate the impact of imperfect verifiers, we add experiments (Appendix A.2) where each EQ call independently fails to find a counterexample with probability  $p \in [0, 0.1]. $ CAPAL remains robust—success typically **80–95%** with essentially flat MQ/EQ budgets—and a simple PAC-style repetition of up to **four EQ calls per round** pushes accuracy back near **100%**, with only a modest constant-factor increase in EQ usage.
>
> ---

---

### Official Review · Reviewer_A92o · 2025-11-03

**Soundness:** 2
**Presentation:** 2
**Contribution:** 3
**Rating:** 2
**Confidence:** 3

**Summary:**

This paper studies active learning of regular languages with membership queries and equivalence queries.This problem regained recent interest due to the usage of LLMs in converting high-level description of regular languages to a deterministic automata, where LLM can serve as an oracle in answering these two types of queries. The paper assumes that responses to membership queries can have persistent noise, while the response to equivalence queries are noiseless. The paper proposes CAPAL algorithm and establishes its correctness guarantees. Experiments show that the proposed algorithm has lower number of queries and a better learning accuracy.

**Strengths:**

- I think the problem motivation is interesting. Converting an informal language description to a formal language allows constructing guardrails of agents, ensuring agent's safety and avoiding unintended consequences in a provable way.

- The experimental results look good to me

**Weaknesses:**

- I am not sure if assuming the equivalence query oracle is noiseless is practical. Such oracle is also implemented using LLM in practice, am I understanding correctly? Then they may hallucinate?

- I wasn't able to follow the presentation of the CAPAL algorithm, perhaps due to me not being an expert in active learning with regular languages. For example, in line 6, how is sa ~ u defined - is it an equivalence relationship defined by the partition in line 5? In line 8, what are the nodes of the decision tree, and what are the splitting criteria for the tree nodes?

Also, it is mentioned in line 307-308 that the membership answers equals to the true label with probability 1-eta with eta < 1/2 -- is this for all the queries? If so, then this seems to be inconsistent with Definition 3.2 (where it is mentioned that for some query, the mode of the oracle's response distribution disagrees with the truth)?

Also, is the label flipping probability assumed to be homogeneous across all queries in theoretical analysis? It seems not the case from the empirical result in Figure 2.

- The paper provides some correctness guarantees of the algorithm (Theorem 1 and 2). It relies on some assumption on the knowledge of the safe noise cap \bar{\eta}. Can this be known, or estimated? Alg. 1 line 2 proposes to estimate \hat{\eta}, but not enough details are given.

- Is there some formal theorem query complexity that can be given? Without that the theory seems incomplete.

**Questions:**

See questions above.  Also, can the authors comment on why in Figure 3, CAPAL's success rate is not always 1 for all epsilon?

---

> ### Author Response · Authors · 2025-11-22
> **Rebuttal to Reviewer A92o**
>
> We thank the reviewer for the constructive comments.  We clarify all points below and will integrate these explanations into the revised paper.  All reference points now direct to the revised version, and all updated content is marked in **blue**.
>
> ---
>
> ### **W1**
>
> The assumption of a *perfect* EQ oracle is the standard one in *40 years of active DFA learning theory* (L\* (Angluin 1987), Rivest–Schapire 1989, TTT (Isberner et al. 2014), etc.), and is precisely what enables strong correctness guarantees. In our target applications, the EQ oracle is not an LLM but a task-specific verifier (e.g., reference DFA, protocol simulator, model checker, or software artifact), so exact counterexamples are practically available whenever verification is cheaper than repeatedly querying an LLM for MQs.
>
> We also explicitly evaluate CAPAL under an *imperfect* EQ oracle that misses counterexamples with probability  $p \le 0.1.$
> CAPAL still attains $\approx 80$–$95$% success with essentially flat MQ/EQ budgets, and a simple PAC-style repetition (up to 4 EQ calls per round) drives accuracy back close to $100$% with only a small constant-factor increase in EQ usage (Appendix A.2).
>
> ---
>
> ### **W2**
>
> **Line 6 (“$sa \sim u$”).**
> Yes, we made the definition more explicit in the revision. This uses the same-state relation defined by our *noise-aware class query*. Section 4 (“Noise-aware class query”, Lines 324–338) defines:
>
> $$
> D(u,v) = \frac{1}{m}\big|\{e \in E : y_e(ue) \neq y_e(ve)\}\big|, \qquad
> u \sim v \iff D(u,v) \le p_0 + \tau,
> $$
>
> where $p_0 = 2\hat\eta(1-\hat\eta)$ and $\tau$ is a Hoeffding tolerance.
>
> **Line 8 (“DT”) — Discrimination tree.**
> DT refers to the *discrimination tree* (Isberner et al. 2014).  Lines 343–350 (p. 7) explain that inner nodes store discriminators $e \in E_\text{core}$, and leaves correspond to hypothesis classes. Consistency violations are repaired by adding a single derived column $a + e_\mathrm{LCA}$, exactly as in TTT. In the revision, we made “DT = discrimination tree’’ explicit at first occurrence.
>
> ---
>
> ### **3. “$1-\eta$” and persistent errors**
>
> The reviewer is correct that the phrase in Sec. 4 may seem to imply a *per-query homogeneous* noise assumption.
> Our **formal** model is in Def. 3.2: each query $q$ has its own correctness probability $p^*(c,q)$.
>
> The scalar $\eta$ in Sec. 4 is the **average mistake rate** (relative to an optimal exact-MAT run) and is used only to set a conservative noise floor  $p_0 = 2\hat\eta(1-\hat\eta)$ for class queries.
>
> Thus:
>
> - We **do not** assume homogeneous flip probabilities.
> - Persistent, query-specific bias is allowed and observed empirically (Fig. 2b, p. 5).
> - We revised the wording in Sec. 4 and tied it directly to Def. 3.2 to avoid confusion.
>
> ---
>
> ### **W3**
>
> Yes. Appendix A.3.2 gives the estimation procedure.  We will reference Appendix A.3.2 explicitly in the main text.
>
> ---
>
> ### **W4**
>
> Our theoretical results (Thm. 1–2) already show:
>
> - soundness under a perfect EQ;
> - finite convergence with high probability under persistent MQ noise;
> - at most $n-1$ discriminators in $E_\text{core}$.
>
> More precisely, for a minimal $n$-state DFA with access diameter $\rho$ and counterexamples of length at most $L_\mathrm{CE}$:
>
> - CAPAL uses  $O(n) \text{ EQ queries},$
> - and  $O(|\Sigma|\,n^2 \rho \;+\; n L_\mathrm{CE}) \text{ MQ queries},$
>
> up to a constant-factor overhead from class queries that depends only on the noise margin and desired confidence.
> Thus the asymptotic query complexity matches that of classical $L^\*$.
>
> We included the full analysis in the Appendix A.4.
>
> ---
>
> ### **Q1**
>
> We impose a global cap of $5\times 10^4$ MQs per task (Line 487).  As $\epsilon \to 0.5$, the class-query test requires more statistical evidence, so some runs exhaust the MQ budget before completion. With a larger MQ budget, these runs also converge.
>
> ---
>
> We hope the above clarifications resolve the concerns and illustrate the robustness and theoretical soundness of our approach.

---

### Official Review · Reviewer_1zKR · 2025-11-06

**Soundness:** 4
**Presentation:** 3
**Contribution:** 3
**Rating:** 8
**Confidence:** 4

**Summary:**

This work considers the problem of learning a deterministic finite automaton from a natural language derived oracle. Specifically, an LLM as a membership oracle to answer yes/no questions about a natural language description of a formal language. This can either be done by answering questions one at a time or converting the description into code, e.g., python. A fundamental problem is that the LLM may make systematic errors in labeling queries. This can result in learning arbitrarily incorrect languages. This is classically addressed by the LEARNANYWAY algorithm, however requires small counterexamples to be made practical. This however is typically not the case as counterexamples are often found via formal analysis or random search/conformance testing.

The contribution of this work is CAPAL algorithm which seems to perform fairly efficiently even in the presence of labeling errors. The key assumption, called pMAT, is that in addition to a natural language membership oracle (with the possibly to err) there is a perfect counter example oracle. The algorithm is relies on a combination of statistical tests and clever usage of a discrimiation tree, ala Kearns and Vazirani.

**Strengths:**

The idea of extracting formal structure from natural language is timely and important. While easy to generate such things in an ad-hoc manner, e.g., code gen, this and related work help provide rigor and guarantees to such procedures. The focus on DFAs is interesting as it is the simplest structure that allows for memory, being the target of decades of learning research.

The techniques are as far as I know novel, and the framing of pMAT makes the underlying assumptions and limitations fairly clear. While clearly strong, see weaknesses, it allows for progress made in understanding this problem.

**Weaknesses:**

1. The primary weakness of this approach is the very strong assumption of an equivalence oracle. While there are a few settings this might make sense, e.g., when an (expensive) verification process exists and is cheap compared to the LLM query required for program translation, it obviously not typically available. In such settings conformance testing and random sampling are much more common, typically yielding PAC like guarantees. This seems like a natural next step, and is explicitly acknowledged in the discussion.

2. Another minor writing point is that, in exposition, this paper leans heavily on existing knowledge of automata learning. While not strictly required, I think without a fairly good automata learning background, a reader will likely bounce off of this work. This is particularly likely with much ICLR community. Nevertheless, I think the paper is understandable.

3. Finally, a quick nit pick about exposition with the L*LM related work. My understanding is that the demonstrations only come into play as a mechanism for refining the language *after* an automata has been guessed using a classic learning algorithm. One should able to drop in CAPAL directly into the L*LM framework and still leverage the demonstration modality right?

**Questions:**

See weakness #3.

Also, do you have a sense how this would compare against SAT based approaches allowing for up to `k` labels being flipped (where `k` is derived from the noise floor?) Presumably, this is more scalable, but on the hand works like L*LM showed that creating distinguishing sequences of the smallest consistent DFAs resulted in less hallucination opportunities.

---

> ### Author Response · Authors · 2025-11-22
> **Rebuttal to Reviewer 1zKR**
>
> We thank the reviewer for the thoughtful and positive assessment of our work and for clearly articulating the motivation and contributions of pMAT and CAPAL. We clarify the specific weaknesses and questions below and will incorporate these changes into the revised paper. All reference points now direct to the revised version, and all updated content is marked in blue.
>
> ---
>
> ### **W1**
>
> The assumption of a *perfect* EQ oracle is the standard one in *40 years of active DFA learning theory* (L\* (Angluin 1987), Rivest–Schapire 1989, TTT (Isberner et al. 2014), etc.), and is precisely what enables strong correctness guarantees. In our target applications, the EQ oracle is a task-specific verifier (e.g., reference DFA, protocol simulator, model checker, or software artifact), so exact counterexamples are practically available whenever verification is cheaper than repeatedly querying an LLM for MQs.
>
> We also explicitly evaluate CAPAL under an *imperfect* EQ oracle that misses counterexamples with probability $p \le 0.1$: CAPAL still attains $\approx 80$–$95$% success with essentially flat MQ/EQ budgets, and a simple PAC-style repetition (up to $4$ EQ calls per round) drives accuracy back close to $100$% with only a small constant-factor increase in EQ usage (Appendix A.2).
>
> ---
>
> ### **W2**
>
> We agree that the paper currently leans on prior familiarity with active DFA learning. To make it more accessible to a broader ICLR audience, we will:
>
> 1. Add a full running example (e.g., the “even number of `a`” DFA) and use it to give a step-by-step explanation of the algorithm block (how `SAMESTATE` is applied, how the discrimination tree is updated, and how the hypothesis is refined after each counterexample), and
> 2. Include a brief primer on the observation-table / discrimination-tree view of DFA learning in the preliminaries.
>
> ---
>
> ### **W3 and Q1**
>
> Yes, the reviewer’s understanding is correct: CAPAL is not a replacement for $L^\*LM$. CAPAL can be dropped into this framework as a direct replacement for the classical DFA learner: demonstrations and the DISS module remain unchanged, and CAPAL simply plays the role of the active learner that issues MQs and processes counterexamples.
>
> ---
>
> ### **Q2**
>
> This comparison is indeed a trade-off: SAT-based learners and other passive learners that allow up to $k$ label flips can be more scalable and require far fewer MQs, but they generally sacrifice correctness because there is no EQ oracle to correct flipped labels; their guarantees are typically PAC-style over a sample rather than exact language identification.
>
> Relative to a fully interactive $L^\*$ loop with many LLM-based MQs, such SAT approaches may offer fewer hallucination opportunities because they ask fewer queries. However, in our program-based oracle setting, CAPAL also keeps hallucinations and LLM cost low—one program synthesis call per task plus cheap program execution—while still enjoying the strong correctness guarantees that come from having a perfect EQ oracle.

---

### Author Response · Authors · 2025-11-26
**General Response**

We thank the reviewers for their constructive feedback. We are encouraged that they view extracting formal structure from natural language as timely and important (1zKR, 6djq), recognize the pMAT formulation as clarifying the assumptions behind LLM-based automata learning (1zKR, dPdj, w6ef), and judge CAPAL to be a principled and effective method for DFA learning under persistent membership noise (1zKR, dPdj, 6djq). Reviewers also found the empirical evaluation thorough and convincing, with several noting the strong performance of the code-based oracle (6djq, w6ef).

Our work addresses the longstanding challenge of active DFA learning with noisy membership queries. This setting follows the classical L* paradigm, where MQs arise from executing a system under learning—and thus are the fragile component—while EQs are answered by formal methods such as conformance testing, automata equivalence checking, or model checking. We formalize this asymmetry via the pMAT framework, pairing a noisy membership oracle (e.g., an LLM) with an accurate verifier-based EQ oracle. CAPAL then uses statistical class queries and a discrimination tree to ensure correctness under persistent, query-specific noise while reducing MQ usage and LLM calls without loss in accuracy.

**EQ-oracle practicality.** The central concern was whether assuming a perfect EQ oracle is realistic. In our applications, the EQ oracle  _is_ a trusted, task-specific verifier—a reference DFA, policy-compliance simulator, model checker, regex/policy engine, or conformance tester—exactly mirroring four decades of practice in L*, Rivest–Schapire, and TTT. To assess robustness when EQs are imperfect, we added new experiments (App. A.2) where EQs independently fail with probability ρ. CAPAL maintains 80–95% success with essentially unchanged budgets, and simple PAC-style repetition (≤4 EQ calls/round) restores success to near 100% with only a small constant-factor overhead. This demonstrates that the EQ assumption is both practically realizable and empirically robust when relaxed.

**Clarity and accessibility (A92o, w6ef).** Some density in the original exposition stemmed from an implicit assumption of reader familiarity with L*-style algorithms. We have substantially revised the text to remove this barrier: expanding explanations of observation-table and discrimination-tree methods (L*, TTT), adding a complete running example (“even number of a”) that illustrates `SAMESTATE` evaluation, discrimination-tree refinement, and counterexample processing, and refining the abstract and introduction to more clearly motivate the pMAT setting and its relevance to applications such as simulator-validated policy extraction.

**Noise model and theory (A92o).** We clarify the heterogeneous noise model, highlight the micro-bootstrap procedure for estimating the noise cap, and summarize the guarantees showing that CAPAL matches classical L* up to small constants.

These updates directly address the main concerns about practicality, clarity, and noise modeling.

---

### Meta-Review · Area_Chair_Qc99 · 2026-01-09

**Summary:**

The paper is about learning deterministic finite automaton (DFA) from a natural language description. The paper develops two types of queries: membership queries (which may have noise) and equivalence queries (which are noiseless). The paper presents the CAPAL algorithm and establishes its theoretical underpinnings.

The paper has had some variance in reviews. It received two 2s and three positive reviews: 6, 6, 8. While on the positive side, the reviewers appreciated the importance of the problem, rigor and guarantees of the methodoloy, potential connections with agent safety and guardrails, and experimentation, there are two main concerns. [1] is this problem setting practical and useful? [2] is the assumption of perfect EQ oracle justifiable in practice?

In response, the authors make three points. [A] in motivation section that add a paragraph giving an example where such a setting may work. Personally, to me that example feels a bit far-fetched/contrived and highly disconnected from the actual experiments.  [B] The authors argue that perfect EQ pracle is a standard assumption in DFA learning literature; an EQ oracle is a task-specific verifier (reference DFA, simulator, conformance tester, regex/policy engine). The authors do not discuss whether such oracles are easy to write and whether their being perfect makes sense in the real world. [C] Finally, the authors add an expt in the appendix where they make this oracle artificially error-prone with independent prob <0.1, and find that their method is robust achieving 85-90% accuracy.

Even after understanding both sides, the acceptance/rejection of this paper becomes a judgment call. If the paper _must_ contribute something realistic and directly useful, then the paper falls short. In fact, I personally am not even sure of the motivating example added in the intro. All experiments are on toy regular languages, taking us back in an older era of AI. Similarly, the robustness experiment assumes independent prob of failure, making that expt a bit contrived too. Finally, what happens in DFA learning literature is not a strong enough reason for accepting the assumption.

On the other hand, if we take the view that this paper builds on an area of inquiry, and adds an interesting algorithm, along with strong therotical foundations on which others may be able to build on, then the paper definitely has something strong to contribute.

The paper will definitely improve significantly with a better motivation for why this problem setting is interesting, and a better set of non-toy experiments for realistic settings... but, should that (especially the latter) be a demand of this paper... probably not.

**Reviewer Concerns:**

see above

**Reviewer Scores:**

The 2s seem harsh ... especially the 2 by A92o. They mostly ask clarification style questions and do not offer a strong justification for a 2. The other 2 by w6ef is reasonable since they clearly highlight their quesitons about motivation and assumption of the paper as the reason for rejection.

My guess is that w6ef would have remained strongly negative. In fact, there is some possibility of the high 8s to go lower maybe to a 6 or so after the response.

---

### Decision · Program_Chairs · 2026-01-26

Accept (Poster)